# A Generalized Bisimulation Metric of State Similarity between Markov Decision Processes: From Theoretical Propositions to Applications

**Zhenyu Tao, Wei Xu, Xiaohu You**
Southeast University
Purple Mountain Laboratories
{zhenyu_tao, wxu, xhyu}@seu.edu.cn

## Abstract

The bisimulation metric (BSM) is a powerful tool for computing state similarities within a Markov decision process (MDP), revealing that states closer in BSM have more similar optimal value functions. While BSM has been successfully utilized in reinforcement learning (RL) for tasks like state representation learning and policy exploration, its application to multiple-MDP scenarios, such as policy transfer, remains challenging. Prior work has attempted to generalize BSM to pairs of MDPs, but a lack of rigorous analysis of its mathematical properties has limited further theoretical progress. In this work, we formally establish a generalized bisimulation metric (GBSM) between pairs of MDPs, which is rigorously proven with the three fundamental properties: GBSM symmetry, inter-MDP triangle inequality, and the distance bound on identical state spaces. Leveraging these properties, we theoretically analyse policy transfer, state aggregation, and sampling-based estimation in MDPs, obtaining explicit bounds that are strictly tighter than those derived from the standard BSM. Additionally, GBSM provides a closed-form sample complexity for estimation, improving upon existing asymptotic results based on BSM. Numerical results validate our theoretical findings and demonstrate the effectiveness of GBSM in multi-MDP scenarios.

## 1 Introduction

Markov decision processes (MDPs) serve as a foundational framework for modeling decision-making problems in Reinforcement Learning (RL) [1]. To enable efficient analysis of MDPs, Ferns et al. [2] proposed the bisimulation metric (BSM) based on the Wasserstein distance, also known as the Kantorovich–Rubinstein metric, to quantify state similarity in a policy-independent manner. BSM provides theoretical guarantees that states closer under this metric exhibit more similar optimal value functions. Meanwhile, BSM is a pseudometric [3] satisfying: (1) Symmetry: $d(s, s') = d(s', s)$, (2) Triangle inequality: $d(s, s') \leq d(s, s'') + d(s'', s')$, and (3) Indiscernibility of identicals: $s = s' \Rightarrow d(s, s') = 0$. These three properties, combined with BSM's measuring capability on optimal value functions, have driven its applications across diverse RL applications. It has been successfully employed in state aggregation [4, 5], representation learning [6, 7], policy exploration [8, 9], goal-conditioned RL [10], safe RL [11], etc.

However, since BSM is inherently defined over a single MDP, its application to theoretical analyses involving multiple MDPs faces notable obstacles. For instance, Phillips [12] applied BSM to policy transfer by constructing a disjoint union of the source and target MDPs' state spaces. While this allows inter-MDP comparisons through BSM, the disjoint union enforces zero transition probabilities between states across the two MDPs. Consequently, this method fixes the total variation distance between their transition probabilities at one, hindering further simplifications and

analysis. It necessitates iterative calculation of distances across the entire state space, leading to prohibitive computational costs in deep RL tasks as noted in [13]. Also, in order to compute state similarities in continuous or large-space discrete MDPs, Ferns et al. [14] proposed a state similarity approximation method through state aggregation and sampling-based estimation. Although they proved the convergence of approximated state similarities to actual ones by leveraging properties of BSM and the Wasserstein distance, their approach only derived a fairly loose approximation error bound and failed to obtain an explicit sample complexity (i.e., the lower bound on the number of samples required to achieve the specified level of accuracy) for the estimation error. Specifically, the estimation error bound (see Eq. 7.1 in [14]) depends on the former aggregation process, resulting in an asymptotic sample complexity rather than a closed-form expression. In addition, for representation learning, Zhang et al. [6] and Kemertas and Aumentado-Armstrong [15] leveraged BSM to establish value function approximation bounds under optimal and non-optimal policies, respectively. However, BSM-based analysis between the original and aggregated MDPs results in loose bounds, particularly with large discount factors.

Several works have attempted to extend the definition of BSM for evaluating similarity between multiple MDPs [16–18]. Notably, when extended to multi-MDP scenarios, this modified version of BSM loses its pseudometric properties, as $s$ and $s'$ in $d(s, s')$ represent states in different MDPs. To the best of our knowledge, prior works have typically extended the single-MDP formulation to two MDPs, without rigorously retaining the metric properties. Specifically, Castro and Precup [16] utilized its evaluation capability on optimal value functions to analyze policy transfer. Due to the lack of metric properties, the derived theoretical performance bound is limited to transferring the optimal policy within the source MDP, as it can only reflect the effect of one-step action rather than the long-term impact of the transferred policy (see Theorem 5 in [16]). While Song et al. [17] successfully employed such a modified BSM in assessing MDP similarities and improving the long-term reward in policy transfer, their investigation focused on empirical validation rather than in-depth theoretical analysis. Furthermore, a comprehensive survey on various state similarity measures between MDPs [18], which highlighted the modified BSM as an effective approach, also emphasized the limited theoretical guarantees in current methodologies. This raises the following two questions:

*Q1. Does the modified BSM possess any metric properties when computing state similarities between multiple MDPs, akin to the pseudometric properties of BSM within a single MDP?*

*Q2. If so, how can these properties facilitate the theoretical analysis involving multiple MDPs?*

To answer Q1, we present a formal definition for the modified BSM in multi-MDP scenarios, which we refer to as generalized BSM (GBSM), and rigorously establish three metric properties that align with the pseudometric properties of BSM. These properties are summarized as (1) GBSM symmetry, (2) inter-MDP triangle inequality, and (3) the distance bound on identical state spaces. To answer Q2, we apply GBSM in the theoretical analyses of policy transfer, state aggregation, and sampling-based estimation of MDPs, yielding explicit bounds for policy transfer performance, aggregation error, and estimation error, respectively. Notably, when the compared MDPs are identical, the error bound of GBSM reduces to the error bound of BSM for a single MDP. We prove that the GBSM-derived bound is strictly tighter than the bound directly obtained from BSM, along with an explicit and closed-form sample complexity for approximation that advances beyond the asymptotic results of [14]. Numerical results corroborate our theoretical findings.

## 2 Background

Before describing the details of our contributions, we give a brief review of the required background in reinforcement learning and the bisimulation metric.

**Reinforcement Learning** We consider an MDP $\langle \mathcal{S}, \mathcal{A}, \mathbb{P}, R, \gamma \rangle$ defined by a finite state space $\mathcal{S}$, a finite action space $\mathcal{A}$, transition probability $\mathbb{P}(\tilde{s}|s, a)$ ($a \in \mathcal{A}$, $\{\tilde{s}, s\} \in \mathcal{S}$, and $\tilde{s}$ denotes the next state), a reward function $R(s, a)$, and a discount factor $\gamma$. Policies $\pi(\cdot|s)$ are mappings from states to distributions over actions, inducing a value function recursively defined by $V^{\pi}(s) := \mathbb{E}_{a \sim \pi(\cdot|s)} \left[ R(s, a) + \gamma \mathbb{E}_{\tilde{s} \sim \mathbb{P}(\cdot|s, a)} \left[ V^{\pi}(\tilde{s}) \right] \right]$. In RL, we are concerned with finding the optimal policy $\pi^* = \arg\max_\pi V^\pi$, which induces the optimal value function denoted by $V^*$.

**Bisimulation Metric** Different definitions of BSM exist in the literature [2, 4, 14]. In this paper, we adopt the formulation from [4], setting the weighting constant to its maximum value $c = \gamma$. The BSM is then defined as:

$$d^\sim(s, s') = \max_a \left\{ |R(s, a) - R(s', a)| + \gamma W_1(\mathbb{P}(\cdot|s, a), \mathbb{P}(\cdot|s', a); d^\sim) \right\}, \forall s, s' \in \mathcal{S}. \quad (1)$$

Here, $W_1$ is the 1-Wasserstein distance, measuring the minimal transportation cost between distributions $\mathbb{P}(\cdot|s, a)$ to $\mathbb{P}(\cdot|s', a)$, with $d^\sim$ as the cost function. Ferns et al. [2] showed that this metric consistently bounds differences in the optimal value function, i.e., $|V^*(s) - V^*(s')| \leq d^\sim(s, s')$.

## 3 Generalized Bisimulation Metric

We now present a formal definition of the proposed GBSM and derive its key metric properties.

**Definition 3.1 (Generalized bisimulation metric).** *Given two MDPs $\mathcal{M}_1 = \langle \mathcal{S}_1, \mathcal{A}, \mathbb{P}_1, R_1, \gamma \rangle$ and $\mathcal{M}_2 = \langle \mathcal{S}_2, \mathcal{A}, \mathbb{P}_2, R_2, \gamma \rangle$, the GBSM between any state $s \in \mathcal{S}_1$ and any state $s' \in \mathcal{S}_2$ is defined as:*

$$d((s, \mathcal{M}_1), (s', \mathcal{M}_2)) = \max_a \left\{ |R_1(s, a) - R_2(s', a)| + \gamma W_1(\mathbb{P}_1(\cdot|s, a), \mathbb{P}_2(\cdot|s', a); d^{1\text{-}2}) \right\}. \quad (2)$$

For notational simplicity, we use $d^{1\text{-}2}(s, s')$ to denote $d((s, \mathcal{M}_1), (s', \mathcal{M}_2))$, where the superscript 1-2 indicates the direction of GBSM from $\mathcal{M}_1$ to $\mathcal{M}_2$. Before proving the existence of GBSM, we first introduce the Wasserstein distance [19], which is defined through the following primal linear program (LP):

$$W_1(P, Q; d) = \min_{\boldsymbol{\lambda}} \sum_{i=1}^{|\mathcal{S}_1|} \sum_{j=1}^{|\mathcal{S}_2|} \lambda_{i,j} d(s_i, s_j),$$
$$\text{subject to } \sum_{j=1}^{|\mathcal{S}_2|} \lambda_{i,j} = P(s_i), \ \forall i ; \sum_{i=1}^{|\mathcal{S}_1|} \lambda_{i,j} = Q(s_j), \ \forall j ; \lambda_{i,j} \geq 0, \ \forall i, j. \quad (3)$$

Here, $P$ and $Q$ are distributions on $\mathcal{S}_1$ and $\mathcal{S}_2$, respectively, and $s_i \in \mathcal{S}_1, s_j \in \mathcal{S}_2$. It represents the minimum transportation cost from $P$ to $Q$ under cost function $d : \mathcal{S}_1 \times \mathcal{S}_2 \to \mathbb{R}_+$, and is equivalent to the following dual LP according to the Kantorovich duality [20]:

$$W_1(P, Q; d) = \max_{\boldsymbol{\mu}, \boldsymbol{\nu}} \sum_{i=1}^{|\mathcal{S}_1|} \mu_i P(s_i) - \sum_{j=1}^{|\mathcal{S}_2|} \nu_j Q(s_j),$$
$$\text{subject to } \mu_i - \nu_j \leq d(s_i, s_j), \ \forall i, j. \quad (4)$$

Then the existence of such a $d^{1\text{-}2}$ satisfying Eq. 2 is established by the following theorem.

**Theorem 3.2 (Existence and convergence of GBSM).** *Let $d_0^{1\text{-}2}$ be a constant zero function and define*

$$d_n^{1\text{-}2}(s, s') = \max_a \left\{ |R_1(s, a) - R_2(s', a)| + \gamma W_1(\mathbb{P}_1(\cdot|s, a), \mathbb{P}_2(\cdot|s', a); d_{n-1}^{1\text{-}2}) \right\}, \ n \in \mathbb{N} \quad (5)$$

*Then $d_n^{1\text{-}2}$ converges to the fixed point $d^{1\text{-}2}$ uniformly with $n \to \infty$. Let $\bar{R} = \max_{s, s', a} |R_1(s, a) - R_2(s', a)|$, and the convergence of $d_n^{1\text{-}2}$ to $d^{1\text{-}2}$ satisfies*

$$d^{1\text{-}2}(s, s') - d_n^{1\text{-}2}(s, s') \leq \gamma^n \bar{R}/(1 - \gamma). \quad (6)$$

*Proof Sketch.* The existence of $d^{1\text{-}2}$ is established through the fixed-point theorem [21] and the definition of the Wasserstein distance, similar to the proof of BSM in [2]. The convergence is proved via the LP in (3) and induction. (See Appendix A for the complete proof.) □

Similar to BSM, which evaluates the state similarity through the optimal value function, GBSM naturally bounds differences in the optimal value function between two MDPs.

**Theorem 3.3 (Optimal value difference bound between MDPs).** *Let $V_1^*$ and $V_2^*$ denote the optimal value functions in $\mathcal{M}_1$ and $\mathcal{M}_2$, respectively. Then the GBSM provides an upper bound for the difference between the optimal values for any state pair $(s, s') \in \mathcal{S}_1 \times \mathcal{S}_2$:*

$$|V_1^*(s) - V_2^*(s')| \leq d^{1\text{-}2}(s, s'). \quad (7)$$

*Proof Sketch.* We first construct a recursive form of the optimal value function by $V^{(n)}(s) = \max_a \left\{ R(s,a) + \gamma \mathbb{E}_{\tilde{s} \sim \mathbb{P}(\cdot|s,a)} \left[ V^{(n-1)}(\tilde{s}) \right] \right\}$, with base case $V^{(0)}(s) = 0$ and $\lim_{n \to \infty} V^{(n)}(s) = V^*$. The proof proceeds by induction on $n$. The key insight is that $\left( V_1^{(n)}(s_k) \right)_{k=1}^{|\mathcal{S}_1|}$ and $\left( V_2^{(n)}(s_k) \right)_{k=1}^{|\mathcal{S}_2|}$ form a feasible, but not necessarily the optimal, solution to the dual LP in (4) for $W_1 \left( \mathbb{P}_1(\cdot|s,a), \mathbb{P}_2(\cdot|s',a); d_n^{1\text{-}2} \right)$. (See Appendix B for the complete proof.) □

Now, we start to establish the three fundamental metric properties of GBSM, which we term GBSM symmetry, inter-MDP triangle inequality, and the distance bound on identical state spaces. These properties are designed to align with pseudometric properties of BSM, including symmetry, triangle inequality, and indiscernibility of identical.

**Theorem 3.4** (**GBSM symmetry**). *Let $d^{1\text{-}2}$ be the GBSM from $\mathcal{M}_1$ to $\mathcal{M}_2$, and $d^{2\text{-}1}$ be the GBSM in the opposite direction, then we have*

$$d^{1\text{-}2}(s,s') = d^{2\text{-}1}(s',s), \ \forall \ (s,s') \in \mathcal{S}_1 \times \mathcal{S}_2. \tag{8}$$

*Proof.* This property can be readily proved through induction. We have $|R_1(s,a) - R_2(s',a)| = |R_2(s',a) - R_1(s,a)|$ for the base case. With the assumption of $d_n^{1\text{-}2}(s,s') = d_n^{2\text{-}1}(s',s)$, we have $W_1 \left( \mathbb{P}_1(\cdot|s,a), \mathbb{P}_2(\cdot|s',a); d_n^{1\text{-}2} \right) = W_1 \left( \mathbb{P}_2(\cdot|s',a), \mathbb{P}_1(\cdot|s,a); d_n^{2\text{-}1} \right)$, and from (5) we have $d_{n+1}^{1\text{-}2}(s,s') = d_{n+1}^{2\text{-}1}(s',s)$. It is therefore concluded that $d_n^{1\text{-}2}(s,s') = d_n^{2\text{-}1}(s',s)$ for all $n \in \mathbb{N}$ and $(s,s') \in \mathcal{S}_1 \times \mathcal{S}_2$. Taking $n \to \infty$ yields the desired result. □

**Theorem 3.5** (**Inter-MDP triangle inequality of GBSM**). *Given MDPs $\mathcal{M}_1 = \langle \mathcal{S}_1, \mathcal{A}, \mathbb{P}_1, R_1, \gamma \rangle$, $\mathcal{M}_2 = \langle \mathcal{S}_2, \mathcal{A}, \mathbb{P}_2, R_2, \gamma \rangle$, and $\mathcal{M}_3 = \langle \mathcal{S}_3, \mathcal{A}, \mathbb{P}_3, R_3, \gamma \rangle$, GBSMs between the three MDPs satisfy*

$$d^{1\text{-}2}(s,s') \leq d^{1\text{-}3}(s,s'') + d^{3\text{-}2}(s'',s'), \ \forall \ (s,s',s'') \in \mathcal{S}_1 \times \mathcal{S}_2 \times \mathcal{S}_3. \tag{9}$$

*Here, the GBSM between any two MDPs can be arbitrarily reversed according to its symmetry.*

*Proof.* First, we need to prove the transitive property of inequality on the Wasserstein distance, that is, the Wasserstein distance between the three distributions follows $W_1(\mathbb{P}^1, \mathbb{P}^2; d^{1\text{-}2}) \leq W_1(\mathbb{P}^1, \mathbb{P}^3; d^{1\text{-}3}) + W_1(\mathbb{P}^3, \mathbb{P}^2; d^{3\text{-}2})$ if (9) holds, where $\mathbb{P}^1$, $\mathbb{P}^2$, and $\mathbb{P}^3$ denote arbitrary distributions on $\mathcal{S}_1$, $\mathcal{S}_2$, and $\mathcal{S}_3$.

Let $(s_i, s_j, s_k) \in \mathcal{S}_1 \times \mathcal{S}_2 \times \mathcal{S}_3$. Define $\boldsymbol{\lambda}^{1,3}$ as the optimal transportation plan for $W_1(\mathbb{P}^1, \mathbb{P}^3; d^{1\text{-}3})$ in primal LP (3), with elements $\lambda_{i,k}^{1,3}$ satisfying $\sum_{k=1}^{|\mathcal{S}_3|} \lambda_{i,k}^{1,3} = \mathbb{P}^1(s_i)$ and $\sum_{i=1}^{|\mathcal{S}_1|} \lambda_{i,k}^{1,3} = \mathbb{P}^3(s_k)$. Similarly define $\boldsymbol{\lambda}^{3,2}$ for $W_1(\mathbb{P}^3, \mathbb{P}^2; d^{3\text{-}2})$ with elements $\lambda_{k,j}^{3,2}$. Construct $\boldsymbol{\lambda}^{1,3,2}$ with elements $\lambda_{i,k,j}^{1,3,2}$ satisfying $\sum_{j=1}^{|\mathcal{S}_2|} \lambda_{i,k,j}^{1,3,2} = \lambda_{i,k}^{1,3}$ and $\sum_{i=1}^{|\mathcal{S}_1|} \lambda_{i,k,j}^{1,3,2} = \lambda_{k,j}^{3,2}$. Such a $\boldsymbol{\lambda}^{1,3,2}$ does exist according to the Gluing Lemma in [22]. Then, note that

$$\sum_{j=1}^{|\mathcal{S}_2|} \sum_{k=1}^{|\mathcal{S}_3|} \lambda_{i,k,j}^{1,3,2} = \sum_{k=1}^{|\mathcal{S}_3|} \lambda_{i,k}^{1,3} = \mathbb{P}^1(s_i),$$

$$\sum_{i=1}^{|\mathcal{S}_1|} \sum_{k=1}^{|\mathcal{S}_3|} \lambda_{i,k,j}^{1,3,2} = \sum_{k=1}^{|\mathcal{S}_3|} \lambda_{k,j}^{3,2} = \mathbb{P}^2(s_j),$$

thus $\sum_{k=1}^{|\mathcal{S}_3|} \boldsymbol{\lambda}^{1,3,2}$ is a feasible, but not necessarily the optimal, solution to the primal LP in (3) for $W_1(\mathbb{P}^1, \mathbb{P}^2; d^{1\text{-}2})$. Consequently, we have

$$
\begin{aligned}
W_1(\mathbb{P}^1, \mathbb{P}^2; d^{1\text{-}2}) &\leq \sum_{i=1}^{|\mathcal{S}_1|} \sum_{j=1}^{|\mathcal{S}_2|} \left( \sum_{k=1}^{|\mathcal{S}_3|} \lambda_{i,k,j}^{1,3,2} \right) d^{1\text{-}2}(s_i, s_j) \\
&\overset{(a)}{\leq} \sum_{i=1}^{|\mathcal{S}_1|} \sum_{j=1}^{|\mathcal{S}_2|} \left( \sum_{k=1}^{|\mathcal{S}_3|} \lambda_{i,k,j}^{1,3,2} \right) \left( d^{1\text{-}3}(s_i, s_k) + d^{3\text{-}2}(s_k, s_j) \right) \\
&= \sum_{i=1}^{|\mathcal{S}_1|} \sum_{k=1}^{|\mathcal{S}_3|} \lambda_{i,k}^{1,3} d^{1\text{-}3}(s_i, s_k) + \sum_{k=1}^{|\mathcal{S}_3|} \sum_{j=1}^{|\mathcal{S}_2|} \lambda_{k,j}^{3,2} d^{3\text{-}2}(s_k, s_j) \\
&= W_1(\mathbb{P}^1, \mathbb{P}^3; d^{1\text{-}3}) + W_1(\mathbb{P}^3, \mathbb{P}^2; d^{3\text{-}2}). \tag{10}
\end{aligned}
$$

Here, step $(a)$ stems from the assumption on $d$. We have now established the transitivity of the inter-MDP triangle inequality on the Wasserstein distance.

Armed with (10), we are ready to prove the inter-MDP triangle inequality of the GBSM through induction. For the base case,

$$d_1^{1\text{-}2}(s, s') \leq \max_a\{|R_1(s, a) - R_3(s'', a)| + |R_3(s'', a) - R_2(s', a)|\}$$
$$\leq \max_a\{|R_1(s, a) - R_3(s'', a)|\} + \max_a\{|R_3(s'', a) - R_2(s', a)|\}$$
$$= d_1^{1\text{-}3}(s, s'') + d_1^{3\text{-}2}(s'', s'), \ \forall (s, s', s'') \in \mathcal{S}_1 \times \mathcal{S}_2 \times \mathcal{S}_3.$$

By the induction hypothesis, we assume that for an arbitrary $n \in \mathbb{N}$,

$$d_n^{1\text{-}2}(s, s') \leq d_n^{1\text{-}3}(s, s'') + d_n^{3\text{-}2}(s'', s'), \ \forall (s, s', s'') \in \mathcal{S}_1 \times \mathcal{S}_2 \times \mathcal{S}_3.$$

The induction follows

$$d_{n+1}^{1\text{-}2}(s, s') \leq \max_a \big\{ \big|R_1(s, a) - R_3(s'', a)\big| + \big|R_3(s'', a) - R_2(s', a)\big|$$
$$+ \gamma W_1\big(\mathbb{P}_1(\cdot|s, a), \mathbb{P}_3(\cdot|s'', a); d_n^{1\text{-}3}\big) + \gamma W_1\big(\mathbb{P}_3(\cdot|s'', a), \mathbb{P}_2(\cdot|s', a); d_n^{3\text{-}2}\big) \big\}$$
$$\leq \max_a \big\{ \big|R_1(s, a) - R_3(s'', a)\big| + \gamma W_1\big(\mathbb{P}_1(\cdot|s, a), \mathbb{P}_3(\cdot|s'', a); d_n^{1\text{-}3}\big) \big\}$$
$$+ \max_a \big\{ \big|R_3(s'', a) - R_2(s', a)\big| + \gamma W_1\big(\mathbb{P}_3(\cdot|s'', a), \mathbb{P}_2(\cdot|s', a); d_n^{3\text{-}2}\big) \big\}$$
$$= d_{n+1}^{1\text{-}3}(s, s'') + d_{n+1}^{3\text{-}2}(s'', s'), \ \forall (s, s', s'') \in \mathcal{S}_1 \times \mathcal{S}_2 \times \mathcal{S}_3.$$

Here, the first inequality follows from (10), i.e., the transitivity of the inequality. Now we have $d_n^{1\text{-}2}(s, s') \leq d_n^{1\text{-}3}(s, s'') + d_n^{3\text{-}2}(s'', s')$ for all $n \in \mathbb{N}$. Taking $n \to \infty$, we establish the inter-MDP triangle inequality of GBSM. $\qquad\square$

Since the identical states only exist within the same state space, we establish the distance bound only when $\mathcal{M}_1$ and $\mathcal{M}_2$ share the same state space $\mathcal{S}$. This property is formulated as follows:

**Theorem 3.6 (Distance bound on identical state spaces).** *When $\mathcal{M}_1$ and $\mathcal{M}_2$ share the same $\mathcal{S}$,*

$$\max_s d^{1\text{-}2}(s, s) \leq \frac{1}{1 - \gamma} \max_{s, a} \Big\{ \big|R_1(s, a) - R_2(s, a)\big| + \frac{\gamma \bar{R}}{1 - \gamma} \text{TV}\big(\mathbb{P}_1(\cdot|s, a), \mathbb{P}_2(\cdot|s, a)\big) \Big\}, \quad (11)$$

*where TV represents the total variation distance defined by* $\text{TV}(P, Q) = \frac{1}{2} \sum_{s \in \mathcal{S}} \big|P(s) - Q(s)\big|$.

*Proof.* Consider a special transportation plan between distributions $P$ and $Q$. This plan preserves all mass shared between $P$ and $Q$, defined as $\min\{P(s), Q(s)\}$ for all $s$. The remaining mass, where $P(s) > Q(s)$, is distributed to states where $P(s) < Q(s)$. Then the total mass to be transported is quantified by the total variation distance, where the transportation cost with the cost function $d$ is bounded by $\max_{s, s'} d(s, s')$. The shared mass is given by $1 - \text{TV}(P, Q)$, with the cost bounded by $\max_s d(s, s)$. While this plan adheres to the definition of Wasserstein distance, it can hardly be optimal. Then we have

$$W_1(P, Q; d) \leq \text{TV}(P, Q) \max_{s, s'} d(s, s') + \big(1 - \text{TV}(P, Q)\big) \max_s d(s, s). \quad (12)$$

According to its recursive definition, GBSM is a mapping bounded by $[0, \bar{R}/(1 - \gamma)]$, then

$$\max_s d^{1\text{-}2}(s, s) \leq \max_{s, a} \big\{ |R_1(s, a) - R_2(s, a)| + \gamma \text{TV}\big(\mathbb{P}_1(\cdot|s, a), \mathbb{P}_2(\cdot|s, a)\big) \max_{s, s'} d^{1\text{-}2}(s, s') \big\}$$
$$+ \gamma \max_{s, a} \big\{ \big(1 - \text{TV}\big(\mathbb{P}_1(\cdot|s, a), \mathbb{P}_2(\cdot|s, a)\big)\big) \max_s d^{1\text{-}2}(s, s) \big\}$$
$$\leq \max_{s, a} \Big\{ |R_1(s, a) - R_2(s, a)| + \frac{\gamma \bar{R}}{1 - \gamma} \text{TV}\big(\mathbb{P}_1(\cdot|s, a), \mathbb{P}_2(\cdot|s, a)\big) \Big\} + \gamma \max_s d^{1\text{-}2}(s, s).$$

Rearranging the inequality yields the desired result. $\qquad\square$

A direct consequence of Theorem 3.6 is that if $\mathcal{M}_1 = \mathcal{M}_2$, where $R_1 = R_2$ and $\mathbb{P}_1 = \mathbb{P}_2$, the right-hand side of the inequality becomes zero. It indicates that

$$d^{1\text{-}1}(s, s) = d((s, \mathcal{M}_1), (s, \mathcal{M}_1)) = 0, \ \forall s \in \mathcal{S}_1, \quad (13)$$

confirming the indiscernibility of identicals of GBSM when the compared objects (state-MDP pairs) are genuinely identical. We denote the maximization term $\max_a\{|R_1(s, a) - R_2(s, a)| + \frac{\gamma \bar{R}}{1 - \gamma} \text{TV}(\mathbb{P}_1(\cdot|s, a), \mathbb{P}_2(\cdot|s, a))\}$ in Theorem 3.6 as $d_{\text{TV}}^{1\text{-}2}(s, s)$ in the following.

We now have a formal definition of GBSM and have rigorously proved the metric properties. Notably, when all compared MDPs are identical, GBSM reduces to the standard BSM. In this case, the three fundamental properties of GBSM reduce to the corresponding pseudometric properties of BSM.

# 4 Applications of GBSM in Multi-MDP Analysis

To demonstrate the effectiveness of GBSM in multi-MDP scenarios, we apply it to theoretical analyses of policy transfer, state aggregation, and sampling-based estimation of MDPs.

## 4.1 Performance Bound of Policy Transfer Using GBSM

Using GBSM, we analyze policy transfer from a source MDP $\mathcal{M}_1$ to a target MDP $\mathcal{M}_2$ and derive a theoretical performance bound for the transferred policy. This bound takes the form of a regret (defined as the expected discounted reward loss incurred by following the transferred policy instead of the optimal one [23]). Specifically, it is a weighted sum of the GBSM between the two MDPs and the regret within the source MDP itself, formulated by the following.

**Theorem 4.1** (**Regret bound on policy transfer**). *Consider transferring a policy $\pi$ from $\mathcal{M}_1$ to $\mathcal{M}_2$. The transferred policy acts as $\pi(\cdot|f(s'))$ for $s' \in \mathcal{S}_2$, where $f : \mathcal{S}_2 \to \mathcal{S}_1$ is a mapping from target states to source states. The regret of $\pi$ in $\mathcal{M}_2$ is bounded by*

$$\max_{s' \in \mathcal{S}_2} |V_2^*(s') - V_2^\pi(s')| \leq \frac{2}{1-\gamma} \max_{s' \in \mathcal{S}_2} d^{1\text{-}2}(f(s'), s') + \frac{1+\gamma}{1-\gamma} \max_{s \in \mathcal{S}_1} |V_1^*(s) - V_1^\pi(s)|. \quad (14)$$

*Proof Sketch.* The proof of (14) is similar to the proof in [12] and is conducted by replacing the BSM by GBSM. (See Appendix C for the complete proof.)  □

Special cases of Theorem 4.1 yield the following refined bounds:

**Corollary 4.2** (**Optimal mapping for policy transfer**). *When $f(s') = \arg\min_{s \in \mathcal{S}_1} d^{1\text{-}2}(s, s')$, $\forall s' \in \mathcal{S}_2$, the bound tightens to:*

$$\max_{s' \in \mathcal{S}_2} |V_2^*(s') - V_2^\pi(s')| \leq \frac{2}{1-\gamma} \max_{s' \in \mathcal{S}_2} \left\{ \min_{s \in \mathcal{S}_1} d^{1\text{-}2}(s, s') \right\} + \frac{1+\gamma}{1-\gamma} \max_{s \in \mathcal{S}_1} |V_1^*(s) - V_1^\pi(s)|. \quad (15)$$

**Corollary 4.3** (**Policy transfer with identical state space**). *When $\mathcal{M}_1$ and $\mathcal{M}_2$ share the same state space $\mathcal{S}$ and $f(s) = s$, we have*

$$\max_{s \in \mathcal{S}} |V_2^*(s) - V_2^\pi(s)| \leq \frac{2}{(1-\gamma)^2} \max_{s \in \mathcal{S}} d_{\text{TV}}^{1\text{-}2}(s, s) + \frac{1+\gamma}{1-\gamma} \max_{s \in \mathcal{S}} |V_1^*(s) - V_1^\pi(s)|. \quad (16)$$

*Proof Sketch.* This corollary utilizes the distance bound on identical state spaces in Theorem 3.6.  □

In contrast to the approach of [12], which constructs a disjoint union state space for analysis, we provide a similar theoretical bound by directly analyzing the relationship between the source and target MDPs. This method avoids a constant total variation distance, thereby enabling simplifications such as the bound based on $d_{\text{TV}}^{1\text{-}2}$, as well as the approximation method in the following section. Meanwhile, calculating BSM on the disjoint union of two MDPs renders a significant computational complexity scaling with $|\mathcal{S}_1 + \mathcal{S}_2|^2$. In contrast, our GBSM is directly computing between $\mathcal{M}_1$ and $\mathcal{M}_2$, with an reduced complexity scaling with $|\mathcal{S}_1| \cdot |\mathcal{S}_2|$.

## 4.2 Approximation Methods and Corresponding Error Bounds

When the state space is extensive and actual transition probabilities are inaccessible, approximation methods are necessary for the efficient computation of state similarities. In the single MDP scenario, Ferns et al. [14] proposed a state similarity approximation (SSA) method based on state aggregation and sampling-based estimation. Let $\mathcal{U} \subseteq \mathcal{S}$ be a set of selected representative states, $[\cdot] : \mathcal{S} \to \mathcal{U}$ an aggregation mapping, $\tilde{\sigma} = \max_{s \in \mathcal{S}} \{d^\sim(s, [s])\}$ the maximum aggregation distance, and $K$ the number of samples used to empirically estimate each transition probability. The SSA error satisfies

$$\max_{s,s'} |d^\sim(s, s') - d_{\tilde{\sigma}, K}^\sim([s], [s'])| \leq \frac{2\tilde{\sigma}(2+\gamma)}{1-\gamma} + \frac{2\gamma}{1-\gamma} \max_{a,s} W_1\left( [\hat{\mathbb{P}}](\cdot|[s], a), [\mathbb{P}](\cdot|[s], a); d^\sim \right). \quad (17)$$

Here, $d_{\tilde{\sigma}, K}^\sim$ denotes the BSM on the approximated MDP, $[\mathbb{P}]$ denotes the transition probability between aggregated states, and $[\hat{\mathbb{P}}]$ represents its empirical counterparts estimated from $K$ samples.

However, the BSM-based aggregation error bound $2\tilde{\sigma}(2+\gamma)/(1-\gamma)$ is fairly loose, while the sample complexity for the estimation error is limited to asymptotic expressions.

Beyond approximating state similarities, it is crucial to quantify the difference between optimal value functions within the original MDPs and their approximated counterparts in aggregated MDPs. Using a BSM-based analysis, Zhang et al. [6] established a value function approximation (VFA) bound on this difference, given by $2\tilde{\sigma}/(1-\gamma)$, but it also suffers from looseness when $\gamma$ becomes large.

To address this, we apply the GBSM to directly compute state similarities between the original MDPs and their aggregated/estimated counterparts. Beyond extending the approach in [14] to the multi-MDP setting, our GBSM-based analysis yields significantly tighter approximation bounds for both SSA and VFA, and provides an explicit and closed-form expression for the sample complexity.

### 4.2.1 State Aggregation

Given the previously defined $\mathcal{S}, \mathcal{U}$, and $[\cdot]$, the aggregated state space $[\mathcal{S}]$ is defined such that the reward function and transition probability of each state are replaced by those of its representative state, given by $R(s,a) = R([s],a)$ and $\mathbb{P}(\cdot|s,a) = \mathbb{P}(\cdot|[s],a)$ for all $s \in \mathcal{S}$. The aggregated transition probability is defined as $[\mathbb{P}](s'|s,a) = \sum_{s'' \in \mathcal{S}, [s'']=s'} \mathbb{P}(s''|s,a)$. Note that $[\mathbb{P}](s'|s,a) = 0$ when $s' \notin \mathcal{U}$. With this construction, we define the aggregated MDP for $\mathcal{M}_1$ as $\mathcal{M}_{[1]} = \langle [\mathcal{S}_1], \mathcal{A}, [\mathbb{P}_1], R_1, \gamma \rangle$. First, we obtain the VFA bound directly from GBSM.

**Theorem 4.4** (**VFA error bound**). *Given MDP $\mathcal{M}_1$ and its aggregated counterpart $\mathcal{M}_{[1]}$, the VFA bound is given by*

$$\max_{s \in \mathcal{S}_1} |V_1^*(s) - V_{[1]}^*(s)| \le \sigma_1 \le \tilde{\sigma}_1/(1-\gamma). \tag{18}$$

*where $\sigma_1 = \max_{s \in \mathcal{S}_1} d^{1\text{-}[1]}(s,s)$ and $\tilde{\sigma}_1 = \max_{s \in \mathcal{S}_1} d^{\sim}(s,[s])$.*

*Proof.* The first inequality is a direct consequence of Theorem 3.3. For the second one, We construct an intermediate MDP defined by $\mathcal{M}_{1_{[\mathcal{S}]}} = \langle [\mathcal{S}_1], \mathcal{A}, \mathbb{P}_1, R_1, \gamma \rangle$ and prove $d^{1\text{-}1_{[\mathcal{S}]}}(s,s) = d^{1\text{-}[1]}(s,s)$ for all $s \in \mathcal{S}_1$ through induction. For the base case, $d_1^{1_{[\mathcal{S}]}\text{-}[1]}(s,s) = \max_a |R_1([s],a) - R_1([s],a)| = 0$. By the induction hypothesis, we assume that $d_n^{1_{[\mathcal{S}]}\text{-}[1]}(s,s) = 0$ for any $n$, then

$$d_{n+1}^{1_{[\mathcal{S}]}\text{-}[1]}(s,s) = \max_a \left\{ |R_1([s],a) - R_1([s],a)| + \gamma W_1(\mathbb{P}(\cdot|[s],a), [\mathbb{P}](\cdot|[s],a); d_n^{1_{[\mathcal{S}]}\text{-}[1]}) \right\}$$

$$\le \gamma \max_a \sum_{\tilde{s} \in \mathcal{S}_1} \mathbb{P}(\tilde{s}|[\tilde{s}],a) d_n^{1_{[\mathcal{S}]}\text{-}[1]}(\tilde{s},[\tilde{s}]).$$

The inequality here follows from a transportation plan that moves the mass from each $\tilde{s}$ to its representative state $[\tilde{s}]$. Note that the reward function and transition probability of each state are the same as its representative states in $\mathcal{M}_{1_{[\mathcal{S}]}}$, thus $d_n^{1_{[\mathcal{S}]}\text{-}[1]}(\tilde{s},[\tilde{s}]) = d_n^{1_{[\mathcal{S}]}\text{-}[1]}([\tilde{s}],[\tilde{s}]) = 0$, and thereby we have $d_{n+1}^{1_{[\mathcal{S}]}\text{-}[1]}(s,s) = 0$. Now we have established $d_n^{1_{[\mathcal{S}]}\text{-}[1]}(s,s) = 0$ for all $n \in \mathbb{N}$ and $s \in \mathcal{S}_1$. Taking $n \to \infty$, we have $d^{1_{[\mathcal{S}]}\text{-}[1]}(s,s) = 0, \forall s \in \mathcal{S}_1$. Using the inter-MDP triangle inequality in Theorem 3.5, we derive $d^{1\text{-}1_{[\mathcal{S}]}}(s,s) = d^{1\text{-}[1]}(s,s)$ for all $s \in \mathcal{S}_1$.

Next, we prove the inequality between $\sigma_1$ and $\tilde{\sigma}_1$. For representative states $s_u \in \mathcal{U}_1 \subseteq \mathcal{S}_1$, we have

$$d^{1\text{-}1_{[\mathcal{S}]}}(s_u, s_u) = \max_a \left\{ |R_1(s_u,a) - R_1(s_u,a)| + \gamma W_1(\mathbb{P}(\cdot|s_u,a), \mathbb{P}(\cdot|s_u,a); d^{1\text{-}1_{[\mathcal{S}]}}) \right\}$$

$$= \gamma \max_a \left\{ W_1(\mathbb{P}(\cdot|s_u,a), \mathbb{P}(\cdot|s_u,a); d^{1\text{-}1_{[\mathcal{S}]}}) \right\}$$

$$\le \gamma \max_a \left\{ \sum_{\tilde{s} \in \mathcal{S}} \mathbb{P}(\tilde{s}|s_u,a) d^{1\text{-}1_{[\mathcal{S}]}}(\tilde{s},\tilde{s}) \right\}$$

$$\le \gamma \max_s d^{1\text{-}1_{[\mathcal{S}]}}(s,s) = \gamma \max_s d^{1\text{-}1_{[\mathcal{S}]}}(s,[s]).$$

Here, the first inequality follows from a straightforward transportation plan that keeps all the mass at its position. The last equality is because $s$ and $[s]$ share the same reward function and transition probability in $\mathcal{M}_{1_{[\mathcal{S}]}}$. Then, according to the inter-MDP triangle inequality in Theorem 3.5, we have

$$d^{1\text{-}1_{[\mathcal{S}]}}(s,[s]) \le d^{1\text{-}1}(s,[s]) + d^{1\text{-}1_{[\mathcal{S}]}}([s],[s]) \le d^{1\text{-}1}(s,[s]) + \gamma \max_s d^{1\text{-}1_{[\mathcal{S}]}}(s,[s])$$

Taking the maximum of both sides, rearranging the inequality, and combining the established $d^{1\text{-}1_{[\mathcal{S}]}}(s,s) = d^{1\text{-}[1]}(s,s)$, we have

$$\sigma_1 = \max_s d^{1\text{-}[1]}(s,s) = \max_s d^{1\text{-}1_{[\mathcal{S}]}}(s,s) = \max_s d^{1\text{-}1_{[\mathcal{S}]}}(s,[s])$$

$$\le \max_s d^{1\text{-}1}(s,[s])/(1-\gamma) = \max_s d^{\sim}(s,[s])/(1-\gamma) = \tilde{\sigma}_1/(1-\gamma),$$

demonstrating significant tightness compared to the BSM-based bound $2\tilde{\sigma}_1/(1-\gamma)$ in [6]. $\qquad\square$

Then the aggregation error bound for SSA is established as follows.

**Theorem 4.5** (**SSA aggregation error bound**). *Given MDPs $\mathcal{M}_1$, $\mathcal{M}_2$ and their aggregated counterparts $\mathcal{M}_{[1]}, \mathcal{M}_{[2]}$, the SSA error bound is given by*

$$\max_{s,s'} |d^{\text{1-2}}(s,s') - d^{[1]\text{-}[2]}(s,s')| \leq \sigma_1 + \sigma_2 \leq (\tilde{\sigma}_1 + \tilde{\sigma}_2)/(1-\gamma). \tag{19}$$

*Proof.* This theorem is easily derived by combining Theorem 3.5 and Theorem 4.4. $\qquad\square$

When the compared MDPs are identical, i.e., $\mathcal{M}_2 = \mathcal{M}_1 = \langle \mathcal{S}, \mathcal{A}, \mathbb{P}, R, \gamma \rangle$, Theorem 4.5 reduces to the aggregation error bound in the single-MDP scenario as

$$\max_{s,s'} |d^{\text{1-1}}(s,s') - d^{[1]\text{-}[1]}(s,s')| \leq 2\sigma_1 \leq 2\tilde{\sigma}_1/(1-\gamma),$$

indicating significant tightness of the GBSM-based bound $2\sigma_1$ compared to the BSM-based one $2\tilde{\sigma}_1(2+\gamma)/(1-\gamma)$ [14].

### 4.2.2 Sampling-based Estimation

To estimate a probability distribution $P$ through statistical sampling, we define the empirical distribution based on $K$ samples as $\hat{P}(x) = \frac{1}{K}\sum_{i=1}^{K} \delta_{X_i}(x)$, where $\{X_1, X_2, \ldots, X_K\}$, are $K$ independent points sampled from $P$ and $\delta$ denotes the Dirac measure at $X_i$ such that $\delta_{X_i}(x) = 1$ if $x = X_i$ and 0 otherwise. Then the empirical MDP for $\mathcal{M}_1$ is constructed by sampling $K$ points for each $\mathbb{P}_1(\cdot|s,a)$, defined by $\mathcal{M}_{\hat{1}} = \langle \mathcal{S}_1, \mathcal{A}, \hat{\mathbb{P}}_1, R_1, \gamma \rangle$. The estimation error bound is derived as follows.

**Theorem 4.6** (**SSA estimation error bound**). *Given MDPs $\mathcal{M}_1$, $\mathcal{M}_2$ and their empirically estimated counterparts $\mathcal{M}_{\hat{1}}, \mathcal{M}_{\hat{2}}$, the SSA error bound is given by*

$$\max_{s,s'} |d^{\text{1-2}}(s,s') - d^{\hat{1}\text{-}\hat{2}}(s,s')| \leq \max_s d^{\text{1-}\hat{1}}(s,s) + \max_{s'} d^{\text{2-}\hat{2}}(s',s'). \tag{20}$$

*To reach an error less than $\epsilon$ with a probability of $1-\alpha$, the sample complexity is given by*

$$K \geq -\ln(\alpha/2)\frac{\gamma^2 \bar{R}^2 |\mathcal{S}_.|^2}{2\epsilon^2(1-\gamma)^4}, \tag{21}$$

*for each state-action pair in $\mathcal{M}_1$ (where $|\mathcal{S}_.| = |\mathcal{S}_1|$) and $\mathcal{M}_2$ (where $|\mathcal{S}_.| = |\mathcal{S}_2|$).*

*Proof.* Inequality (20) is easily obtained from Theorem 3.5. In terms of the sample complexity, we derive the following using Theorem 3.6

$$\max_{s,s'} |d^{\text{1-2}}(s,s') - d^{\hat{1}\text{-}\hat{2}}(s,s')|$$

$$\leq \frac{\gamma\bar{R}}{(1-\gamma)^2}\left(\max_{s,a} \text{TV}\big(\mathbb{P}_1(\cdot|s,a),\hat{\mathbb{P}}_1(\cdot|s,a)\big) + \max_{s',a} \text{TV}\big(\mathbb{P}_2(\cdot|s',a),\hat{\mathbb{P}}_2(\cdot|s',a)\big)\right)$$

$$= \frac{\gamma\bar{R}}{2(1-\gamma)^2}\left(\max_{s,a}\left\{\sum_{\tilde{s}\in\mathcal{S}_1}|\mathbb{P}_1(\tilde{s}|s,a)-\hat{\mathbb{P}}_1(\tilde{s}|s,a)|\right\} + \max_{s',a}\left\{\sum_{\tilde{s}\in\mathcal{S}_2}|\mathbb{P}_2(\tilde{s}|s',a)-\hat{\mathbb{P}}_2(\tilde{s}|s',a)|\right\}\right).$$

To ensure the estimation error remains below $\epsilon$, we require $|\mathbb{P}_1(\tilde{s}|s,a) - \hat{\mathbb{P}}_1(\tilde{s}|s,a)| \leq \frac{\epsilon(1-\gamma)^2}{\gamma\bar{R}|\mathcal{S}_1|}$ and $|\mathbb{P}_2(\tilde{s}|s,a) - \hat{\mathbb{P}}_2(\tilde{s}|s,a)| \leq \frac{\epsilon(1-\gamma)^2}{\gamma\bar{R}|\mathcal{S}_2|}$. Next, by applying the Hoeffding's inequality [24] that is defined by $\Pr\{|\hat{P}(s) - P(s)| \geq \epsilon\} \leq 2\text{e}^{-2K\epsilon^2}$, we derive the desired sample complexity. $\qquad\square$

When the compared MDPs are identical, the estimation SSA bound in Theorem 4.6 reduces to $2\max_s d^{\text{1-}\hat{1}}(s,s)$ for BSM. We now prove the tightness of this new sampling error bound compared to the existing bound $\frac{2\gamma}{1-\gamma}\max_{a,s} W_1(\hat{\mathbb{P}}(\cdot|s,a),\mathbb{P}(\cdot|s,a);d^{\sim})$ in [14]. According to the transitive property of inequality on the Wasserstein distance defined in (10), we have

$$d^{\text{1-}\hat{1}}(s,s) = \gamma W_1(\mathbb{P}(\cdot|s,a),\hat{\mathbb{P}}(\cdot|s,a);d^{\text{1-}\hat{1}})$$

$$\leq \gamma W_1(\mathbb{P}(\cdot|s,a),\hat{\mathbb{P}}(\cdot|s,a),;d^{\text{1-1}}) + \gamma W_1(\hat{\mathbb{P}}(\cdot|s,a),\hat{\mathbb{P}}(\cdot|s,a);d^{\text{1-}\hat{1}})$$

$$\leq \gamma W_1(\hat{\mathbb{P}}(\cdot|s,a),\mathbb{P}(\cdot|s,a);d^{\text{1-1}}) + \gamma\max_s d^{\text{1-}\hat{1}}(s,s).$$

Taking the maximum of both sides and rearranging the inequality yields

$$2\max_s d^{1\text{-}\hat{1}}(s,s) \leq 2\gamma \max_{a,s} W_1(\hat{\mathbb{P}}(\cdot|s,a), \mathbb{P}(\cdot|s,a); d^{1\text{-}1})/(1-\gamma).$$

Since $d^{1\text{-}1} \triangleq d^{\sim}$, we have now proved the tightness of the new sampling error bound compared to the one derived from BSM in [14].

Furthermore, in case the approximation combines both state aggregation and sampling-based estimation, where the approximated MDP is defined as $\mathcal{M}_{[\hat{1}]} = \langle [\mathcal{S}_1], \mathcal{A}, [\hat{\mathbb{P}}_1], R_1, \gamma \rangle$, we have

$$\max_{s,s'} \left| d^{1\text{-}1}(s,s') - d^{[\hat{1}]\text{-}[\hat{1}]}(s,s') \right| \leq 2\max_s d^{1\text{-}[\hat{1}]}(s,s) \leq 2\max_s d^{1\text{-}[1]}(s,s) + 2\max_s d^{[1]\text{-}[\hat{1}]}(s,s)$$

via the inter-MDP triangle inequality. It enables a decoupled analysis of error, and thus results in an explicit and closed-formed sample complexity, i.e., $K \geq -\ln(\alpha/2)\frac{\gamma^2 \bar{R}^2 |\mathcal{U}|^2}{2\epsilon^2(1-\gamma)^4}$ for an error below $\epsilon$ with probability of $1-\alpha$, where $\mathcal{U}$ is the set of representative states.

## 5   Extensions to BSM variants

The proposed GBSM framework is readily extendable to numerous variants of BSM to enhance its applicability, such as lax BSM [25] and on-policy BSM [26].

Lax GBSM enables the computation of state similarities between MDPs with different action spaces. To relax GBSM to lax GBSM, we first adapt (2) to

$$\delta(d)((s,a),(s',a')) = |R_1(s,a) - R_2(s',a')| + \gamma W_1(\mathbb{P}_1(\cdot|s,a), \mathbb{P}_2(\cdot|s',a'); d), \qquad (22)$$

and define the lax function as $F_{\text{lax}}(d|s,s') = H(X_s, X'_{s'}; \delta(d))$, where $X_s = \{(s,a)|a \in \mathcal{A}_1\}$, $X'_{s'} = \{(s',a')|a' \in \mathcal{A}_2\}$, and $H$ is the Hausdorff metric. Iterating from $d^{1\text{-}2}_{\text{lax},0}(s,s') = 0$ and $d^{1\text{-}2}_{\text{lax},n+1} = F_{\text{lax}}(d^{1\text{-}2}_{\text{lax},n}|s,s')$, $d^{1\text{-}2}_{\text{lax},n}$ converges to a similar fixed point $d^{1\text{-}2}_{\text{lax}}$ that satisfies $|V_1^*(s) - V_2^*(s')| \leq d^{1\text{-}2}_{\text{lax}}(s,s')$ in Theorem 3.3. Next, the symmetry (Theorem 3.4) and triangle inequality (Theorem 3.5) can be readily established for $d^{1\text{-}2}_{\text{lax}}$. For MDPs sharing same $\mathcal{S}$ and $\mathcal{A}$, we have $d^{1\text{-}2}_{\text{lax}} \leq d^{1\text{-}2} \leq d^{1\text{-}2}_{\text{TV}}/(1-\gamma)$ (Theorem 3.6). Since these fundamental metric properties hold, the bounds for state aggregation (Theorem 4.5) and estimation (Theorem 4.6) also follow directly. For policy transfer, a similar regret bound (replacing $d^{1\text{-}2}$ in Theorem 4.1 by $d^{1\text{-}2}_{\text{lax}}$) can be established by defining an additional action mapping $g : \mathcal{A}_1 \to \mathcal{A}_2$ for transfer. Due to the introduction of max-min term via Hausdorff metric, the lax GBSM-based transfer bound requires an assumption on this action mapping, i.e., $g(a) = \arg\min_{a'} \delta((f(s'),a),(s',a'); d^{1\text{-}2}_{\text{lax}})$ for each $s'$ and $a$. See Appendix D for the proof.

On-policy GBSM computes state similarities between MDPs under non-optimal policies. To achieve this, we rewrite (2) to

$$d^{1\text{-}2}_{\pi}(s,s') = |R_1^{\pi}(s) - R_2^{\pi}(s')| + \gamma W_1(\mathbb{P}_1^{\pi}(\cdot|s), \mathbb{P}_2^{\pi}(\cdot|s'); d^{1\text{-}2}), \qquad (23)$$

where $R_1^{\pi}(s) = \sum_a \pi(a|s)R_1(s,a)$ and $\mathbb{P}_1^{\pi}(\tilde{s}|s) = \sum_a \pi(a|s)\mathbb{P}_1(\tilde{s}|s,a), \forall \tilde{s} \in \mathcal{S}_1$ represent the expected reward and transition probabilities for a non-optimal policy $\pi$ in $\mathcal{M}_1$, with corresponding terms $R_2^{\pi}$ and $\mathbb{P}_2^{\pi}$ defined similarly for $\mathcal{M}_2$. Our theoretical properties are also preserved in this setting: the value difference bound in Theorem 3.3 now applies to the on-policy value function by $|V_1^{\pi}(s) - V_2^{\pi}(s')| \leq d^{1\text{-}2}_{\pi}(s,s')$. Then metric properties Theorem 3.4 and 3.5 follow directly, and $d^{1\text{-}2}_{\pi}(s,s')$ is bounded by an on-policy TV-based metric $d^{1\text{-}2}_{\text{TV},\pi}(s,s') = \{|R_1^{\pi}(s) - R_2^{\pi}(s')| + \frac{\gamma \bar{R}}{1-\gamma}\text{TV}(\mathbb{P}_1^{\pi}(\cdot|s), \mathbb{P}_2^{\pi}(\cdot|s))\}$ as the Theorem 3.6 for on-policy GBSM. As a direct consequence, we have $\max_s |V_1^{\pi}(s) - V_{[1]}^{\pi}(s)| \leq \max_s d^{1\text{-}[1]}_{\pi}(s,s) \leq \max_s \tilde{d}_{\pi}(s,[s])/(1-\gamma)$, a tighter bound for VFA with non-optimal policy compared with the existing result $2\tilde{d}_{\pi}(s,[s])/(1-\gamma)$ in [15]. See Appendix E for the proof.

## 6   Numerical Results

In this section, we empirically validate the theoretical results derived from GBSM. To this end, we construct MDPs with randomly generated reward functions and transition probabilities, along with their aggregated and estimated counterparts. Specifically, we use random Garnet MDPs with

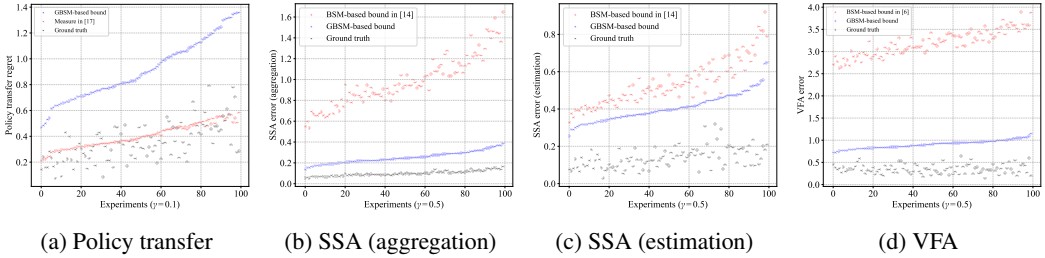

| (a) Policy transfer | (b) SSA (aggregation) | (c) SSA (estimation) | (d) VFA |

Figure 1: Experiments on random Garnet MDPs.

$|\mathcal{S}| = 20, |\mathcal{A}| = 5$, and a 50% branching factor. In the aggregated MDPs, the reward functions and transition probabilities for half of the states are replaced by those of their representative states, while the estimated MDPs are established by introducing a Gaussian noise with a standard deviation ranging from 0.1 to 0.3 to the transition probabilities.

To demonstrate the application of policy transfer between MDPs, we calculate the bound in Theorem 4.1 and the existing measure between MDPs in [17], and calculate the ground-truth regret by computing the precise value functions under a tabular Q-learning setting. Then, we calculate the aggregation and estimation SSA bounds using BSM and GBSM, respectively. The BSM-based SSA bounds are computed via (17). Since the estimation error bound in (17) depends on the aggregation process, we decouple the two for clearer analysis. Specifically, the BSM-based aggregation SSA bound is given by $2\tilde{\sigma}_1(2+\gamma)/(1-\gamma)$, and the estimation SSA bound is $\frac{2\gamma}{1-\gamma} \max_{a,s} W_1(\hat{\mathbb{P}}(\cdot|s,a), \mathbb{P}(\cdot|s,a); d^{\sim})$ [14]. The GBSM-based SSA bounds follow from Theorem 4.5 (aggregation) and Theorem 4.6 (estimation). For VFA bounds comparison, we employ the GBSM-based bound in Theorem 4.4 and compare with the BSM-based bound $2\tilde{\sigma}_1/(1-\gamma)$ in [6]. We also compare them with the ground-truth error values to assess their tightness.

We conduct 100 independent experiments for each $\gamma \in \{0.1, 0.2, \ldots, 0.9\}$. The x-axis represents the experiment index of 100 independent trials, while the y-axis plots the values of ground-truth error and (G)BSM-based bounds in each trial. Figure 1a shows that the empirical metric in [17] fails to bound the transfer regret, while our GBSM-based bound is consistently effective. In terms of the SSA and VFA error, as depicted in Figure 1b, 1c, and 1d, the bounds based on GBSM are significantly tighter than those derived from BSM, which corroborates our theoretical findings and highlights the effectiveness of GBSM in multi-MDP analysis. Complete results are provided in Appendix F.

## 7    Conslusion

**Application and limitation** The first application is the sim-to-real policy transfer, where GBSM can be calculated between the simulated MDP and real-world MDP to predict transferred performance and serve as a metric for improving the simulation environment. Meanwhile, the approximation methods could be employed to address the inaccessibility of precise transition probabilities and reward functions in the real world. Another potential application is in multi-task RL, where GBSM can coordinate policy optimization across different MDPs, cluster similar tasks for efficient training, and mitigate gradient interference issues. The limitation mainly lies in the discounted reward formulation in GBSM. In real-world tasks, the goal is typically to maximize the long-term average reward. However, most of the theoretical results in this paper are divided by $1-\gamma$, and taking $\gamma$ to 1 would yield an infinite result. Investigating metrics tailored for the average-reward MDP is an important and promising direction for future research.

**Discussion** In this paper, we have formally introduced GBSM and established its fundamental theoretical properties, including GBSM symmetry, inter-MDP triangle inequality, and distance bound on identical state spaces. Leveraging these properties, we provide tighter bounds for policy transfer, state aggregation, and sampling-based estimation of MDPs, compared to the ones derived from BSM. To our knowledge, this is the first rigorous theoretical investigation of GBSM beyond simple definitional adaptation. We believe this work introduces a valuable new tool for multi-MDP analysis.

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

# A Proof of Theorem 3.2

This section provides a detailed proof of the existence and convergence of GBSM.

## A.1 The Existence of $d^{1\text{-}2}$

To prove the existence of $d^{1\text{-}2}$, we introduce the Knaster-Tarski fixed-point theorem. Let $(\mathcal{X}, \preceq)$ denote a partial order, which means certain pairs of elements within the set $\mathcal{X}$ are comparable under the homogeneous relation $\preceq$ [21]. If this partial order has least upper bounds and greatest lower bounds for its arbitrary subsets, it is called a complete lattice. The Knaster-Tarski fixed-point theorem asserts that for a continuous function on a complete lattice, the iterative application of this function to the least element of the lattice converges to a fixed point $\bar{x}$, which satisfies $\bar{x} = f(\bar{x})$. Formally, the theorem is stated as follows.

**Lemma A.1** (**Knaster-Tarski fixed-point theorem** [21]). *If the partial order $(\mathcal{X}, \preceq)$ is a complete lattice and $f : \mathcal{X} \to \mathcal{X}$ is a continuous function. Then, $f$ has a least fixed point, given by*

$$\mathrm{fix}(f) = \sqcup_{n \in \mathbb{N}} f^{(n)}(x_0), \tag{24}$$

*where $x_0$ is the least element of $\mathcal{X}$, $\sqcup$ denotes the least upper bound, $f^{(n)}(x_0) = f(f^{(n-1)}(x_0))$, and $f^{(1)}(x_0) = f(x_0)$. Here, the continuity of $f$ is defined such that for any increasing sequence $\{x_n\}$ in $\mathcal{X}$, it satisfies*

$$f\left(\sqcup_{n \in \mathbb{N}} \{x_n\}\right) = \sqcup_{n \in \mathbb{N}} \{f(x_n)\}. \tag{25}$$

Let $\mathcal{D}$ denote the set of all cost functions, which are defined as maps that satisfy $\mathcal{S}_1 \times \mathcal{S}_2 \to [0, \frac{\bar{R}}{1-\gamma}]$. Equip $\mathcal{D}$ with the usual pointwise ordering: Consider two cost function, say $d$ and $d' \in \mathcal{D}$, denote $d \leq d'$ if and only if $d(s, s') \leq d'(s, s')$ for any $s \in \mathcal{S}_1$ and $s' \in \mathcal{S}_2$. Then $\mathcal{D}$ forms a complete lattice with the least element $d_0^{1\text{-}2}$, i.e., the constant zero function. Given $s$ and $s'$, we regard the recursive definition in (5) as a function of $d$ and accordingly define $F : \mathcal{D} \to \mathcal{D}$ by

$$F(d \mid s, s') = \max_a \left\{ \left| R_1(s, a) - R_2(s', a) \right| + \gamma W_1 \big( \mathbb{P}_1(\cdot | s, a), \mathbb{P}_2(\cdot | s', a); d \big) \right\}. \tag{26}$$

Utilizing the Knaster-Tarski fixed-point theorem, the existence of $d^{1\text{-}2}$ is achieved if the continuity of $F$ holds on $\mathcal{D}$.

We first prove the continuity of the second term in $F$. Define $F_{W_1} : \mathcal{D} \to \mathcal{D}$ by

$$F_{W_1}(d \mid s, s') = W_1 \big( \mathbb{P}_1(\cdot | s, a), \mathbb{P}_2(\cdot | s', a); d \big). \tag{27}$$

**Lemma A.2.** *$F_{W_1}$ is continuous on $\mathcal{D}$.*

*Proof.* We follow the definition of continuity defined in Lemma A.1. Let $s_i \in \mathcal{S}_1$ and $s_j \in \mathcal{S}_2$. Regard $F_{W_1}(s_i, s_j; d)$ as a function of $d$. Without loss of generality, we denote probability distributions $\{\mathbb{P}_1(\cdot | s_i, a), \mathbb{P}_2(\cdot | s_j, a)\}$ as $\{P, Q\}$ for brevity, and let $\rho \leq \rho'$, $\{\rho, \rho'\} \in \mathcal{D}$. Considering the optimal solution $\{\boldsymbol{\mu}, \boldsymbol{\nu}\}$ for $W_1(P, Q; \rho)$ in the dual LP in (4), we have

$$\mu_i - \nu_j \leq \rho(s_i, s_j) \leq \rho'(s_i, s_j), \ \forall i, j, \tag{28}$$

which is derived from the pointwise ordering in $\mathcal{D}$. Here, for the other $W_1(P, Q; \rho')$, $\{\boldsymbol{\mu}, \boldsymbol{\nu}\}$ is a feasible, though not necessarily optimal, solution to the dual LP in (4). Thus, we have

$$W_1(P, Q; \rho) = \sum_{i=1}^{|\mathcal{S}_1|} \mu_i P(s_i) - \sum_{j=1}^{|\mathcal{S}_2|} \nu_j Q(s_j)$$
$$\leq W_1(P, Q; \rho'), \ \forall \rho \leq \rho'. \tag{29}$$

By such a monotonicity, we have $W_1(P, Q; \rho) \leq W_1(P, Q; \sqcup_{n \in \mathbb{N}} \{\rho_n\})$, $\forall \rho \in \{\rho_n\}$ for any increasing sequence $\{\rho_n\}$ on $\mathcal{D}$. This further implies that $\sqcup_{n \in \mathbb{N}} \{W_1(P, Q; \rho_n)\} \leq W_1(P, Q; \sqcup_{n \in \mathbb{N}} \{\rho_n\})$.

We use the primal LP for the other side. Let $\boldsymbol{\lambda}^n$ denote the optimal solution in (3) for $W_1(P, Q; \rho_n)$, which also satisfies the conditions for $W_1(P, Q; \sqcup_{n \in \mathbb{N}} \{\rho_n\})$. Define $\epsilon_{i,j}^n = \sqcup_{n \in \mathbb{N}} \{\rho_n\}(s_i, s_j) -$

$\rho_n(s_i, s_j)$, then $\epsilon_{i,j}^n \geq 0$ and $\lim_{n\to\infty} \epsilon_{i,j}^n = 0$ due to the monotonicity of the increasing sequence of $\{\rho_n\}$. Then, we have

$$
\begin{aligned}
W_1(P, Q; \sqcup_{n\in\mathbb{N}}\{\rho_n\}) &\overset{(a)}{\leq} \sum_{i=1}^{|\mathcal{S}_1|}\sum_{j=1}^{|\mathcal{S}_2|} \lambda_{i,j}^n \cdot \sqcup_{n\in\mathbb{N}}\{\rho_n\}(s_i, s_j) \\
&= \sum_{i=1}^{|\mathcal{S}_1|}\sum_{j=1}^{|\mathcal{S}_2|} \lambda_{i,j}^n \rho_n(s_i, s_j) + \sum_{i=1}^{|\mathcal{S}_1|}\sum_{j=1}^{|\mathcal{S}_2|} \lambda_{i,j}^n \epsilon_{i,j}^n \\
&= W_1(P, Q; \rho_n) + \sum_{i,j=1}^{|\mathcal{S}|} \lambda_{i,j}^n \epsilon_{i,j}^n \\
&\leq \sqcup_{n\in\mathbb{N}}\{W_1(P, Q; \rho_n)\} + \max_{i,j}\{\epsilon_{i,j}^n\}.
\end{aligned}
\tag{30}
$$

Here, step $(a)$ follows from the fact that $\boldsymbol{\lambda}^n$ is the optimal solution for $W_1(P, Q; \rho_n)$ rather than $W_1(P, Q; \sqcup_{n\in\mathbb{N}}\{\rho_n\})$. Taking $n \to \infty$, we have $\sqcup_{n\in\mathbb{N}}\{W_1(P, Q; \rho_n)\} \geq W_1(P, Q; \sqcup_{n\in\mathbb{N}}\{\rho_n\})$. Following from the above two inequalities from both directions, it is readily to get $\sqcup_{n\in\mathbb{N}}\{W_1(P, Q; \rho_n)\} = W_1(P, Q; \sqcup_{n\in\mathbb{N}}\{\rho_n\})$. Thus, for any $i$ and $j$,

$$
\begin{aligned}
F_{W_1}\left(\sqcup_{n\in\mathbb{N}}\{\rho_n\} \,|\, s_i, s_j\right) &= W_1\left(\mathbb{P}_1(\cdot|s_i, a), \mathbb{P}_2(\cdot|s_j, a); \sqcup_{n\in\mathbb{N}}\{\rho_n\}\right) \\
&= \sqcup_{n\in\mathbb{N}}\left\{W_1(\mathbb{P}_1(\cdot|s_i, a), \mathbb{P}_2(\cdot|s_j, a); \rho_n)\right\} \\
&= \sqcup_{n\in\mathbb{N}}\left\{F_{W_1}\left(\rho_n \,|\, s_i, s_j\right)\right\}.
\end{aligned}
\tag{31}
$$

Now that the continuity of $F_{W_1}$ in (27) on $\mathcal{D}$ is established. $\square$

Armed with Lemma A.2, we are ready to establish the continuity of $F$ as follows.

**Lemma A.3.** *$F$ is continuous on $\mathcal{D}$.*

*Proof.* Considering an arbitrary increasing sequence $\{\rho_n\}$ on $\mathcal{D}$, for any $i$ and $j$, we have

$$
\begin{aligned}
&F\left(\sqcup_{n\in\mathbb{N}}\{\rho_n\} \,|\, s_i, s_j\right) \\
&= \max_a \left\{|R_1(s_i, a) - R_2(s_j, a)| + \gamma W_1\left(\mathbb{P}_1(\cdot|s_i, a), \mathbb{P}_2(\cdot|s_j, a); \sqcup_{n\in\mathbb{N}}\{\rho_n\}\right)\right\} \\
&= \max_a \left\{|R_1(s_i, a) - R_2(s_j, a)| + \gamma \sqcup_{n\in\mathbb{N}}\left\{W_1\left(\mathbb{P}_1(\cdot|s_i, a), \mathbb{P}_2(\cdot|s_j, a); \rho_n\right)\right\}\right\} \\
&= \sqcup_{n\in\mathbb{N}}\left\{\max_a \left\{|R_1(s_i, a) - R_2(s_j, a)| + \gamma W_1\left(\mathbb{P}_1(\cdot|s_i, a), \mathbb{P}_2(\cdot|s_j, a); \rho_n\right)\right\}\right\} \\
&= \sqcup_{n\in\mathbb{N}}\left\{F(\rho_n \,|\, s_i, s_j)\right\}.
\end{aligned}
\tag{32}
$$

$\square$

Now that the existence of $d^{1\text{-}2}$ is established by using Lemma A.1 and Lemma A.3.

## A.2 The Convergence of $d_n^{1\text{-}2}$ to $d^{1\text{-}2}$

Due to the continuity of $F$ and using the induction starting from $d_0^{1\text{-}2} \leq d_1^{1\text{-}2}$, $\{d_n^{1\text{-}2}\}$ forms an increasing sequence on $\mathcal{D}$. Given that $d^{1\text{-}2} = \sqcup_{n\in\mathbb{N}} F^{(n)}(d_0^{1\text{-}2})$, we have $d^{1\text{-}2} \geq d_n^{1\text{-}2}$ for any $n$. Also,

$$
\begin{aligned}
d^{1\text{-}2}(s_i, s_j) &= \max_a \left\{|R_1(s_i, a) - R_2(s_j, a)| + \gamma W_1\left(\mathbb{P}_1(\cdot|s_i, a), \mathbb{P}_2(\cdot|s_j, a); d^{1\text{-}2}\right)\right\} \\
&\leq \bar{R} + \gamma \max_{i,j}\{d^{1\text{-}2}(s_i, s_j)\} \\
\Rightarrow \max_{i,j}\{d^{1\text{-}2}(s_i, s_j)\} &\leq \bar{R}/(1 - \gamma), \, \forall\, (s_i, s_j) \in \mathcal{S}_1 \times \mathcal{S}_2.
\end{aligned}
\tag{33}
$$

We begin with a simple inequality for the Wasserstein distance before proving the convergence of GBSM. Let $\boldsymbol{\lambda}^n$ denote the optimal solution for $W_1(P, Q; d_n^{1\text{-}2})$, then for any $d_n^{1\text{-}2}$

$$
\begin{aligned}
W_1\left(P, Q; d^{1\text{-}2}\right) &\leq \sum_{i=1}^{|\mathcal{S}_1|}\sum_{j=1}^{|\mathcal{S}_2|} \lambda_{i,j}^n d^{1\text{-}2}(s_i, s_j) \\
&= \sum_{i=1}^{|\mathcal{S}_1|}\sum_{j=1}^{|\mathcal{S}_2|} \lambda_{i,j}^n \left(d^{1\text{-}2}(s_i, s_j) - d_n^{1\text{-}2}(s_i, s_j) + d_n^{1\text{-}2}(s_i, s_j)\right) \\
&\leq \max_{i,j}\left\{d^{1\text{-}2}(s_i, s_j) - d_n^{1\text{-}2}(s_i, s_j)\right\} + W_1\left(P, Q; d_n^{1\text{-}2}\right).
\end{aligned}
\tag{34}
$$

The first inequality follows from the fact that $\boldsymbol{\lambda}^n$ is the optimal solution for $W_1(P, Q; d_n^{1\text{-}2})$ rather than $W_1(P, Q; d^{1\text{-}2})$.

Now we employ the mathematical induction. For the base case, we have

$$
\begin{aligned}
& d^{1\text{-}2}(s, s') - d_1^{1\text{-}2}(s, s') \\
&= \max_a \left\{ |R_1(s, a) - R_2(s', a)| + \gamma W_1\left(\mathbb{P}_1(\cdot|s, a), \mathbb{P}_2(\cdot|s', a); d^{1\text{-}2}\right) \right\} \\
&\quad - \max_a \left\{ |R_1(s, a) - R_2(s', a)| \right\} \\
&\leq \max_a \left\{ |R_1(s, a) - R_2(s', a)| \right\} + \gamma \max_a \left\{ W_1\left(\mathbb{P}_1(\cdot|s, a), \mathbb{P}_2(\cdot|s', a); d^{1\text{-}2}\right) \right\} \\
&\quad - \max_a \left\{ |R_1(s, a) - R_2(s', a)| \right\} \\
&= \gamma \max_a \left\{ W_1\left(\mathbb{P}_1(\cdot|s, a), \mathbb{P}_2(\cdot|s', a); d^{1\text{-}2}\right) \right\} \\
&\leq \gamma \max_{s, s'} \left\{ d^{1\text{-}2}(s, s') \right\} = \gamma \bar{R}/(1 - \gamma), \ \forall \, (s, s') \in \mathcal{S}_1 \times \mathcal{S}_2.
\end{aligned}
\tag{35}
$$

By the induction hypothesis, we assume that for an arbitrary $n$,

$$
d^{1\text{-}2}(s, s') - d_n^{1\text{-}2}(s, s') \leq \gamma^n \bar{R}/(1 - \gamma), \ \forall \, (s, s') \in \mathcal{S}_1 \times \mathcal{S}_2.
\tag{36}
$$

Then we have

$$
\begin{aligned}
& d^{1\text{-}2}(s, s') - d_{n+1}^{1\text{-}2}(s, s') \\
&= \max_a \left\{ |R_1(s, a) - R_2(s', a)| + \gamma W_1\left(\mathbb{P}_1(\cdot|s, a), \mathbb{P}_2(\cdot|s', a); d^{1\text{-}2}\right) \right\} \\
&\quad - \max_a \left\{ |R_1(s, a) - R_2(s', a)| + \gamma W_1\left(\mathbb{P}_1(\cdot|s, a), \mathbb{P}_2(\cdot|s', a); d_n^{1\text{-}2}\right) \right\} \\
&\leq \max_a \left\{ \left( |R_1(s, a) - R_2(s', a)| + \gamma W_1\left(\mathbb{P}_1(\cdot|s, a), \mathbb{P}_2(\cdot|s', a); d^{1\text{-}2}\right) \right) \right. \\
&\qquad \left. - \left( |R_1(s, a) - R_2(s', a)| + \gamma W_1\left(\mathbb{P}_1(\cdot|s, a), \mathbb{P}_2(\cdot|s', a); d_n^{1\text{-}2}\right) \right) \right\} \\
&= \gamma \max_a \left\{ W_1\left(\mathbb{P}_1(\cdot|s, a), \mathbb{P}_2(\cdot|s', a); d^{1\text{-}2}\right) - W_1\left(\mathbb{P}_1(\cdot|s, a), \mathbb{P}_2(\cdot|s', a); d_n^{1\text{-}2}\right) \right\} \\
&\overset{(a)}{\leq} \gamma \max_a \left\{ \max_{s, s'} \left\{ d^{1\text{-}2}(s, s') - d_n^{1\text{-}2}(s, s') \right\} + W_1\left(\mathbb{P}_1(\cdot|s, a), \mathbb{P}_2(\cdot|s', a); d_n^{1\text{-}2}\right) \right. \\
&\qquad \left. - W_1\left(\mathbb{P}_1(\cdot|s, a), \mathbb{P}_2(\cdot|s', a); d_n^{1\text{-}2}\right) \right\} \\
&= \gamma \max_{s, s'} \left\{ d^{1\text{-}2}(s, s') - d_n^{1\text{-}2}(s, s') \right\} \leq \gamma^{n+1} \bar{R}/(1 - \gamma), \ \forall \, (s, s') \in \mathcal{S}_1 \times \mathcal{S}_2.
\end{aligned}
\tag{37}
$$

Here, step $(a)$ uses (34). Following from (35)-(37), $d^{1\text{-}2}(s, s') - d_n^{1\text{-}2}(s, s') \leq \gamma^n \bar{R}/(1 - \gamma)$ holds for all $n \in \mathbb{N}$.

## B  Proof of Theorem 3.3

This proves the optimal value difference bound between MDPs by induction. For the base case, we have

$$
\begin{aligned}
|V_1^{(1)}(s_i) - V_2^{(1)}(s_j)| &= |\max_a R_1(s_i, a) - \max_a R_2(s_j, a)| \\
&\leq \max_a |R_1(s_i, a) - R_2(s_j, a)| \\
&= d_1^{1\text{-}2}(s_i, s_j), \ \forall \, (s_i, s_j) \in \mathcal{S}_1 \times \mathcal{S}_2.
\end{aligned}
\tag{38}
$$

By the induction hypothesis, we assume that for an arbitrary $n$,

$$
\begin{aligned}
V_1^{(n)}(s_i) - V_2^{(n)}(s_j) &\leq |V_1^{(n)}(s_i) - V_2^{(n)}(s_j)| \\
&\leq d_n^{1\text{-}2}(s_i, s_j), \ \forall \, (s_i, s_j) \in \mathcal{S}_1 \times \mathcal{S}_2.
\end{aligned}
\tag{39}
$$

Then the induction follows

$$\left|V_1^{(n+1)}(s_i) - V_2^{(n+1)}(s_j)\right|$$

$$= \left|\max_a \left\{R_1(s_i, a) + \gamma \sum_{k=1}^{|\mathcal{S}_1|} \mathbb{P}_1(s_k|s_i, a)V_1^{(n)}(s_k)\right\}\right.$$

$$\left. - \max_a \left\{R_2(s_j, a) + \gamma \sum_{k=1}^{|\mathcal{S}_2|} \mathbb{P}_2(s_k|s_j, a)V_2^{(n)}(s_k)\right\}\right|$$

$$\leq \max_a \left\{\left|\left(R_1(s_i, a) + \gamma \sum_{k=1}^{|\mathcal{S}_1|} \mathbb{P}(s_k|s_i, a)V_1^{(n)}(s_k)\right)\right.\right.$$

$$\left.\left. - \left(R_2(s_j, a) + \gamma \sum_{k=1}^{|\mathcal{S}_2|} \mathbb{P}_2(s_k|s_j, a)V_2^{(n)}(s_k)\right)\right|\right\}$$

$$\leq \max_a \left\{\left|R_1(s_i, a) - R_2(s_j, a)\right| + \gamma\left|\sum_{k=1}^{|\mathcal{S}_1|} \mathbb{P}_1(s_k|s_i, a)V_1^{(n)}(s_k) - \sum_{k=1}^{|\mathcal{S}_2|} \mathbb{P}_2(s_k|s_j, a)V_2^{(n)}(s_k)\right|\right\}$$

$$\overset{(a)}{\leq} \max_a \left\{|R_1(s_i, a) - R_2(s_j, a)| + \gamma W_1\left(\mathbb{P}_1(\cdot|s_i, a), \mathbb{P}_2(\cdot|s_j, a); d_n^{1\text{-}2}\right)\right\}$$

$$= d_{n+1}^{1\text{-}2}(s_i, s_j), \ \forall (s_i, s_j) \in \mathcal{S}_1 \times \mathcal{S}_2. \tag{40}$$

Here, steps $(a)$ follows from the fact that $\left(V_1^{(n)}(s_k)\right)_{k=1}^{|\mathcal{S}_1|}$ and $\left(V_2^{(n)}(s_k)\right)_{k=1}^{|\mathcal{S}_2|}$ form a feasible, but not necessarily the optimal, solution to the dual LP in (4) for $W_1\left(\mathbb{P}_1(\cdot|s_i, a), \mathbb{P}_2(\cdot|s_j, a); d_n^{1\text{-}2}\right)$.

Now from (38)-(40), we have $|V_1^{(n)}(s) - V_2^{(n)}(s')| \leq d_n^{1\text{-}2}(s, s'), \ \forall (s, s') \in \mathcal{S}_1 \times \mathcal{S}_2, \ \forall n \in \mathbb{N}$. Taking $n \to \infty$ yields the desired result.

## C  Proof of Theorem 4.1

This section provides a detailed proof of the regret bound for policy $\pi$ transferred from $\mathcal{M}_1$ to $\mathcal{M}_2$. By the triangle inequality, for any state $s_j \in \mathcal{S}_2$ and $s_i = f(s_j) \in \mathcal{S}_1$, we have

$$|V_2^*(s_j) - V_2^\pi(s_j)| \leq |V_2^*(s_j) - V_1^*(s_i)| + |V_1^*(s_i) - V_1^\pi(s_i)| + |V_1^\pi(s_i) - V_2^\pi(s_j)|. \tag{41}$$

Within the right-hand side of this inequality, the first summation term $|V_2^*(s_j) - V_1^*(s_i)|$ is upper bounded by $d^{1\text{-}2}(s_i, s_j)$ according to Theorem 3.3, and $|V_1^*(s_i) - V_1^\pi(s_i)|$ is upper bounded by $\max_{s \in \mathcal{S}_1} |V_1^*(s) - V_1^\pi(s)|$. For the last term, we have

$$|V_1^\pi(s_i) - V_2^\pi(s_j)|$$

$$= \left|\sum_{a=1}^{|\mathcal{A}|} \pi(a|s_i)\left(R_1(s_i, a) + \gamma \sum_{k=1}^{|\mathcal{S}_1|} \mathbb{P}_1(s_k|s_i, a)V_1^\pi(s_k)\right)\right.$$

$$\left. - \sum_{a=1}^{|\mathcal{A}|} \pi(a|f(s_j))\left(R_2(s_j, a) + \gamma \sum_{k=1}^{|\mathcal{S}_2|} \mathbb{P}_2(s_k|s_j, a)V_2^\pi(s_k)\right)\right|$$

$$\leq \sum_{a=1}^{|\mathcal{A}|} \pi(a|s_i)\left(\left|R_1(s_i, a) - R_2(s_j, a)\right| + \gamma\left|\sum_{k=1}^{|\mathcal{S}_1|} \mathbb{P}_1(s_k|s_i, a)V_1^\pi(s_k) - \sum_{k=1}^{|\mathcal{S}_2|} \mathbb{P}_2(s_k|s_j, a)V_2^\pi(s_k)\right|\right)$$

$$\leq \sum_{a=1}^{|\mathcal{A}|} \pi(a|s_i)\left(\left|R_1(s_i, a) - R_2(s_j, a)\right| + \gamma\left|\sum_{k=1}^{|\mathcal{S}_1|} \mathbb{P}_1(s_k|s_i, a)V_1^*(s_k) - \sum_{k=1}^{|\mathcal{S}_2|} \mathbb{P}_2(s_k|s_j, a)V_2^*(s_k)\right|\right)$$

$$+ \gamma \sum_{a=1}^{|\mathcal{A}|} \pi(a|s_i)\left|\sum_{k=1}^{|\mathcal{S}_1|} \mathbb{P}_1(s_k|s_i, a)(V_1^*(s_k) - V_1^\pi(s_k))\right|$$

$$+ \gamma \sum_{a=1}^{|\mathcal{A}|} \pi(a|s_i)\left|\sum_{k=1}^{|\mathcal{S}_2|} \mathbb{P}_2(s_k|s_j, a)(V_2^*(s_k) - V_2^\pi(s_k))\right|$$

$$\leq \max_a \left\{ \left| R_1(s_i, a) - R_2(s_j, a) \right| + \gamma \left| \sum_{k=1}^{|\mathcal{S}_1|} \mathbb{P}_1(s_k|s_i, a)V_1^*(s_k) - \sum_{k=1}^{|\mathcal{S}_2|} (\mathbb{P}(s_k|s_j, a)V_2^*(s_k) \right| \right\}$$

$$+ \gamma \max_{s \in \mathcal{S}_1} |V_1^*(s) - V_1^\pi(s)| + \gamma \max_{s \in \mathcal{S}_2} |V_2^*(s) - V_2^\pi(s)|$$

$$\overset{(a)}{\leq} \max_a \left\{ |R_1(s_i, a) - R_2(s_j, a)| + \gamma W_1 \left( \mathbb{P}_1(\cdot|s_i, a), \mathbb{P}_2(\cdot|s_j, a); d^{1\text{-}2} \right) \right\}$$

$$+ \gamma \max_{s \in \mathcal{S}_1} |V_1^*(s) - V_1^\pi(s)| + \gamma \max_{s \in \mathcal{S}_2} |V_2^*(s) - V_2^\pi(s)|$$

$$= d^{1\text{-}2}(s_i, s_j) + \gamma \max_{s \in \mathcal{S}_1} |V_1^*(s) - V_1^\pi(s)| + \gamma \max_{s \in \mathcal{S}_2} |V_2^*(s) - V_2^\pi(s)|. \tag{42}$$

Here, step $(a)$ stems from the fact that, according to Theorem 3.3, $\left( V_1^*(s_k) \right)_{k=1}^{|\mathcal{S}_1|}$ and $\left( V_2^*(s_k) \right)_{k=1}^{|\mathcal{S}_2|}$ form a feasible, but not necessarily the optimal, solution to the dual LP in (4) for $W_1 \left( \mathbb{P}_1(\cdot|s_i, a), \mathbb{P}_2(\cdot|s_j, a); d^{1\text{-}2} \right)$. Combining the above inequalities on all three summation terms in (41) and taking the maximum of both sides, we have

$$\max_{s' \in \mathcal{S}_2} |V_2^*(s') - V_2^\pi(s')| \leq \underbrace{\max_{s' \in \mathcal{S}_2} d^{1\text{-}2}(f(s'), s')}_{\text{1st term}} + \underbrace{\max_{s \in \mathcal{S}_1} |V_1^*(s) - V_1^\pi(s)|}_{\text{2nd term}}$$

$$+ \underbrace{\max_{s' \in \mathcal{S}_2} d^{1\text{-}2}(f(s'), s') + \gamma \max_{s \in \mathcal{S}_1} \left| V_1^*(s) - V_1^\pi(s) \right| + \gamma \max_{s' \in \mathcal{S}_2} \left| V_2^*(s') - V_2^\pi(s') \right|}_{\text{3rd term}}$$

$$\leq 2 \max_{s \in \mathcal{S}_2} d^{1\text{-}2}(f(s), s) + (1 + \gamma) \max_{s \in \mathcal{S}_1} |V_1^*(s) - V_1^\pi(s)| + \gamma \max_{s' \in \mathcal{S}_2} \left| V_2^*(s') - V_2^\pi(s') \right|. \tag{43}$$

Rearranging the inequality yields the desired result.

## D  Proofs of Lax GBSM-based transfer bound

To prove the policy transfer bound based on lax GBSM, apart from the state mapping $f : \mathcal{S}_2 \to \mathcal{S}_1$, we need to define an additional action mapping $g : \mathcal{A}_1 \to \mathcal{A}_2$ for policy transfer between different action spaces. Then, we have

$$|V_1^\pi(s_i) - V_2^\pi(s_j)|$$

$$= \left| \sum_{a=1}^{|\mathcal{A}_1|} \pi(a|s_i)(R_1(s_i, a) + \gamma \sum_{k=1}^{|\mathcal{S}_1|} \mathbb{P}_1(s_k|s_i, a)V_1^\pi(s_k)) \right.$$

$$\left. - \sum_{a=1}^{|\mathcal{A}_1|} \pi(a|f(s_j))(R_2(s_j, g(a)) + \gamma \sum_{k=1}^{|\mathcal{S}_2|} \mathbb{P}_2(s_k|s_j, g(a))V_2^\pi(s_k)) \right|$$

$$\leq \sum_{a=1}^{|\mathcal{A}_1|} \left( \pi(a|s_i)(|R_1(s_i, a) - R_2(s_j, g(a))| \right.$$

$$\left. + \gamma| \sum_{k=1}^{|\mathcal{S}_1|} \mathbb{P}_1(s_k|s_i, a)V_1^\pi(s_k) - \sum_{k=1}^{|\mathcal{S}_2|} \mathbb{P}_2(s_k|s_j, g(a))V_2^\pi(s_k)|) \right)$$

$$\leq \sum_{a=1}^{|\mathcal{A}_1|} \left( \pi(a|s_i)(|R_1(s_i, a) - R_2(s_j, g(a))| \right.$$

$$\left. + \gamma| \sum_{k=1}^{|\mathcal{S}_1|} \mathbb{P}_1(s_k|s_i, a)V_1^*(s_k) - \sum_{k=1}^{|\mathcal{S}_2|} \mathbb{P}_2(s_k|s_j, g(a))V_2^*(s_k)|) \right)$$

$$+ \gamma \sum_{a=1}^{|\mathcal{A}_1|} \pi(a|s_i)| \sum_{k=1}^{|\mathcal{S}_1|} \mathbb{P}_1(s_k|s_i, a)(V_1^*(s_k) - V_1^\pi(s_k))|$$

$$+ \gamma \sum_{a=1}^{|\mathcal{A}_1|} \pi(a|s_i)| \sum_{k=1}^{|\mathcal{S}_2|} \mathbb{P}_2(s_k|s_j, g(a))(V_2^*(s_k) - V_2^\pi(s_k))|$$

$$\leq \max_{a \in \mathcal{A}_1} \left\{ |R_1(s_i, a) - R_2(s_j, g(a))| \right.$$

$$\left. + \gamma| \sum_{k=1}^{|\mathcal{S}_1|} \mathbb{P}_1(s_k|s_i, a)V_1^*(s_k)) - \sum_{k=1}^{|\mathcal{S}_2|} (\mathbb{P}(s_k|s_j, g(a))V_2^*(s_k)| \right\}$$

$$+ \gamma \max_{s \in \mathcal{S}_1} |V_1^*(s) - V_1^\pi(s)| + \gamma \max_{s \in \mathcal{S}_2} |V_2^*(s) - V_2^\pi(s)|$$

$$\overset{(a)}{\leq} \max_{a \in \mathcal{A}_1} \min_{a' \in \mathcal{A}_2} \{|R_1(s_i, a) - R_2(s_j, a')| + \gamma W_1(\mathbb{P}_1(\cdot|s_i, a), \mathbb{P}_2(\cdot|s_j, a'); d_{\text{lax}}^{1\text{-}2})\}$$

$$+ \gamma \max_{s \in \mathcal{S}_1} |V_1^*(s) - V_1^\pi(s)| + \gamma \max_{s \in \mathcal{S}_2} |V_2^*(s) - V_2^\pi(s)|$$

$$\leq H(\delta(d_{\text{lax}}^{1\text{-}2}))(X_{s_i}, X'_{s_j}) + \gamma \max_{s \in \mathcal{S}_1} |V_1^*(s) - V_1^\pi(s)| + \gamma \max_{s \in \mathcal{S}_2} |V_2^*(s) - V_2^\pi(s)|$$

$$= d_{\text{lax}}^{1\text{-}2}(s_i, s_j) + \gamma \max_{s \in \mathcal{S}_1} |V_1^*(s) - V_1^\pi(s)| + \gamma \max_{s \in \mathcal{S}_2} |V_2^*(s) - V_2^\pi(s)|$$

Due to the introduction of max-min term via Hausdorff metric, step (a) requires that action mapping satisfies $g(a) = \arg\min_{a'} \delta((f(s'), a), (s', a'); d_{\text{lax}}^{1\text{-}2})$ for each $s'$ and $a$.

# E Proofs of on-policy GBSM-based Bound

**Theorem E.1** (**On-policy GBSM optimal value difference bound**). *Let $V_1^\pi$ and $V_2^\pi$ denote the value functions with policy $\pi$ in $\mathcal{M}_1$ and $\mathcal{M}_2$, respectively. Then on-policy GBSM provides an upper bound for the difference between the value functions for any state pair $(s, s') \in \mathcal{S}_1 \times \mathcal{S}_2$:*

$$|V_1^\pi(s) - V_2^\pi(s')| \leq d_\pi^{1\text{-}2}(s, s'). \tag{44}$$

*Proof.* For the base case, we have

$$|V_1^{\pi,(0)}(s_i) - V_2^{\pi,(0)}(s_j)| = d_{\pi,0}^{1\text{-}2}(s_i, s_j) = 0, \ \forall \, (s_i, s_j) \in \mathcal{S}_1 \times \mathcal{S}_2. \tag{45}$$

By the induction hypothesis, we assume that for an arbitrary $n$,

$$|V_1^{\pi,(n)}(s_i) - V_2^{\pi,(n)}(s_j)| \leq d_{\pi,n}^{1\text{-}2}(s_i, s_j), \ \forall \, (s_i, s_j) \in \mathcal{S}_1 \times \mathcal{S}_2. \tag{46}$$

The induction follows

$$\left|V_1^{\pi,(n+1)}(s_i) - V_2^{\pi,(n+1)}(s_j)\right|$$

$$= \left|\left\{R_1^\pi(s_i) + \gamma \sum_{k=1}^{|\mathcal{S}_1|} \mathbb{P}_1^\pi(s_k|s_i) V_1^{\pi,(n)}(s_k)\right\} - \left\{R_2^\pi(s_j) + \gamma \sum_{k=1}^{|\mathcal{S}_2|} \mathbb{P}_2^\pi(s_k|s_j) V_2^{\pi,(n)}(s_k)\right\}\right|$$

$$\leq \left\{\left|R_1^\pi(s_i) - R_2^\pi(s_j)\right| + \gamma \left|\sum_{k=1}^{|\mathcal{S}_1|} \mathbb{P}_1^\pi(s_k|s_i) V_2^{\pi,(n)}(s_k) - \sum_{k=1}^{|\mathcal{S}_2|} \mathbb{P}_2^\pi(s_k|s_j) V_2^{\pi,(n)}(s_k)\right|\right\}$$

$$\leq \left\{\left|R_1^\pi(s_i) - R_2^\pi(s_j)\right| + \gamma W_1\left(\mathbb{P}_1^\pi(\cdot|s_i), \mathbb{P}_2^\pi(\cdot|s_j); d_n^{1\text{-}2}\right)\right\}$$

$$= d_{n+1}^{1\text{-}2}(s_i, s_j), \ \forall \, (s_i, s_j) \in \mathcal{S}_1 \times \mathcal{S}_2. \tag{47}$$

Taking $n \to \infty$ yields the desired result. $\qquad\square$

**Theorem E.2** (**VFA error bound with non-optimal policy**).

$$\max_s |V_1^\pi(s) - V_{[1]}^\pi(s)| \leq \max_s d_\pi^{1\text{-}[1]}(s, s) \leq \max_s \tilde{d}_\pi(s, [s])/(1 - \gamma) \tag{48}$$

*Proof.* The first inequality follows directly from Theorem E.1, while the second is established using a derivation analogous to the proof of Theorem 4.4. $\qquad\square$

# F Additional Numerical Results

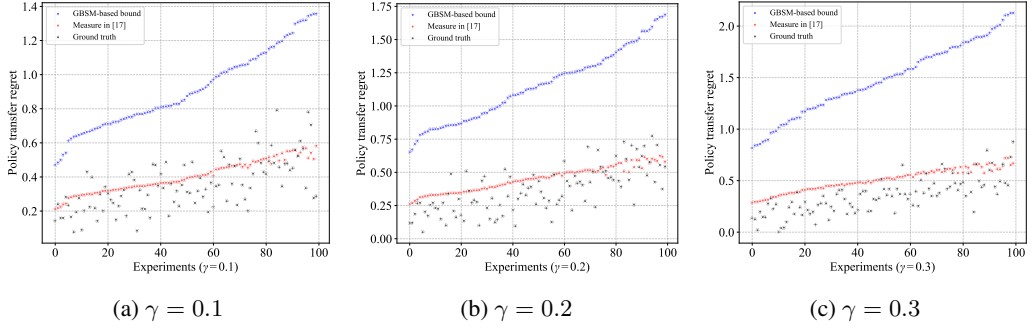

Figure 2: Experiments on random Garnet MDPs (policy transfer, $\gamma = 0.1$ to $0.3$).

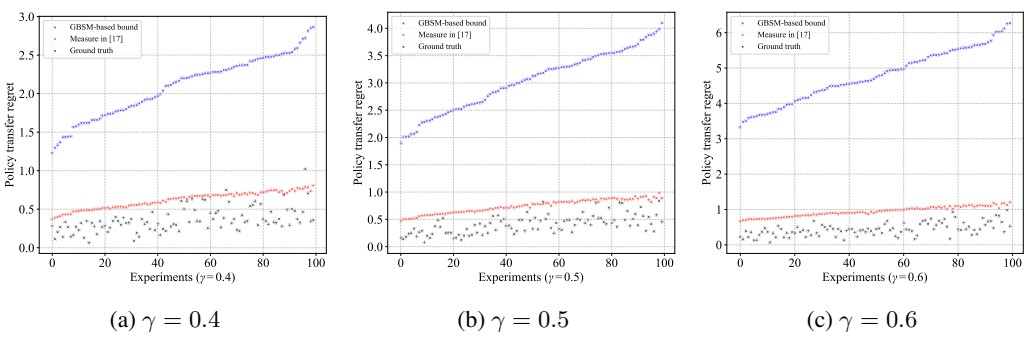

Figure 3: Experiments on random Garnet MDPs (policy transfer, $\gamma = 0.4$ to $0.6$).

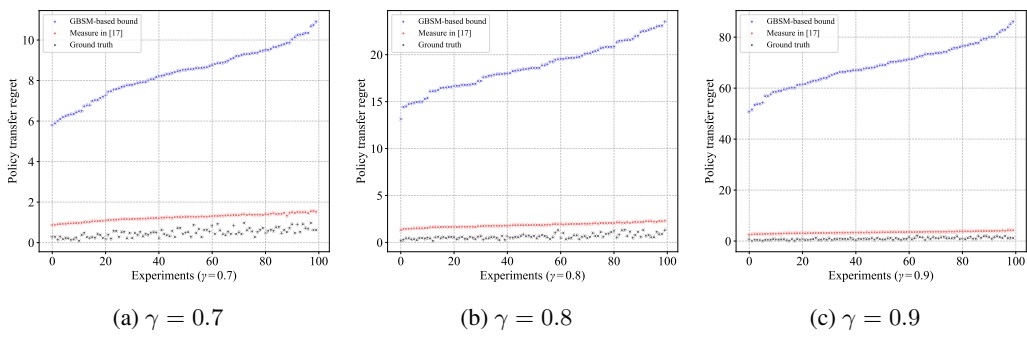

Figure 4: Experiments on random Garnet MDPs (policy transfer, $\gamma = 0.7$ to $0.9$).

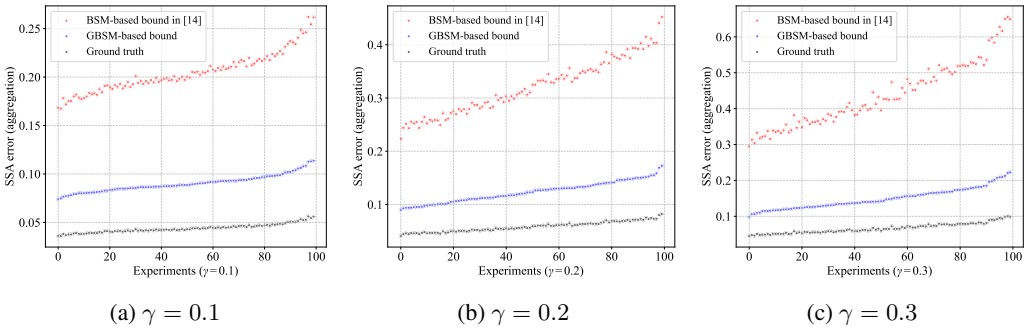

Figure 5: Experiments on random Garnet MDPs (SSA with aggregation, $\gamma = 0.1$ to $0.3$).

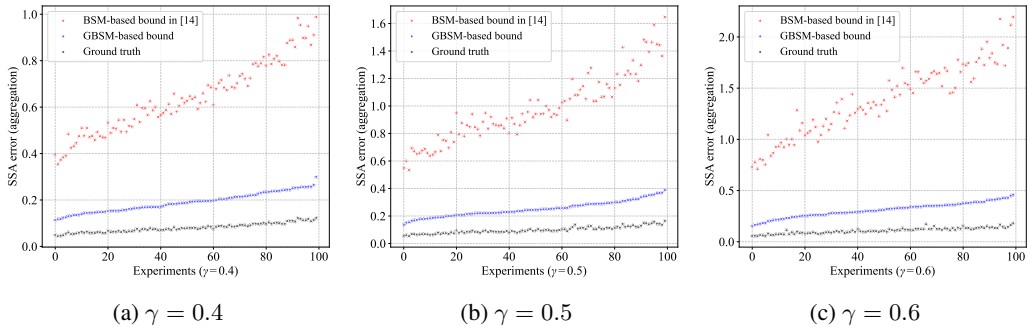

(a) $\gamma = 0.4$  (b) $\gamma = 0.5$  (c) $\gamma = 0.6$

Figure 6: Experiments on random Garnet MDPs (SSA with aggregation, $\gamma = 0.4$ to $0.6$).

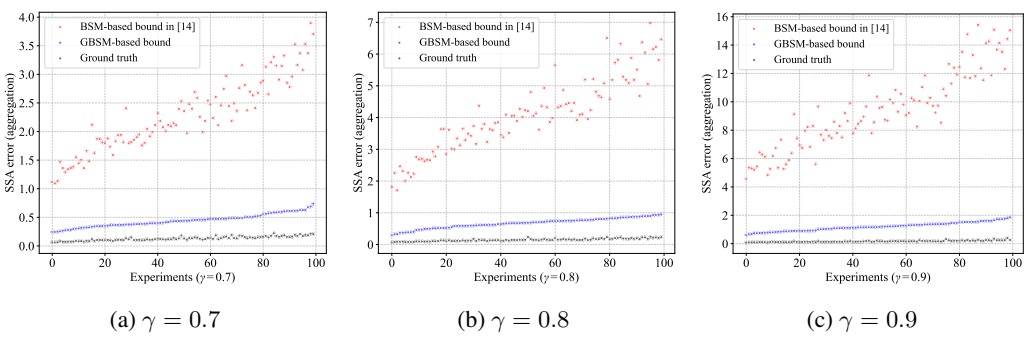

(a) $\gamma = 0.7$  (b) $\gamma = 0.8$  (c) $\gamma = 0.9$

Figure 7: Experiments on random Garnet MDPs (SSA aggregation, $\gamma = 0.7$ to $0.9$).

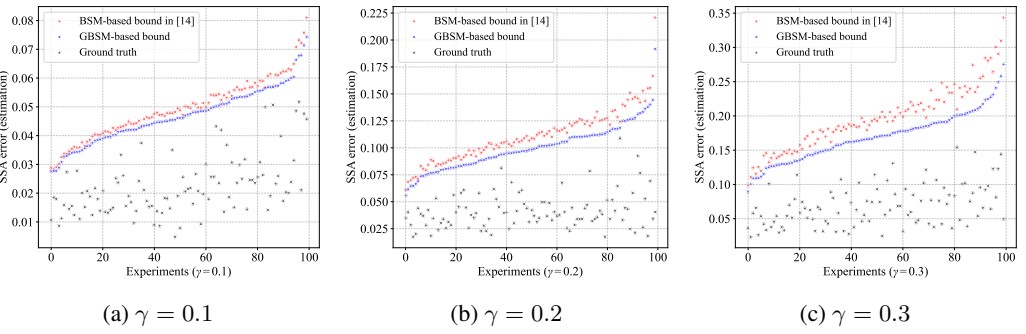

(a) $\gamma = 0.1$  (b) $\gamma = 0.2$  (c) $\gamma = 0.3$

Figure 8: Experiments on random Garnet MDPs (SSA with estimation, $\gamma = 0.1$ to $0.3$).

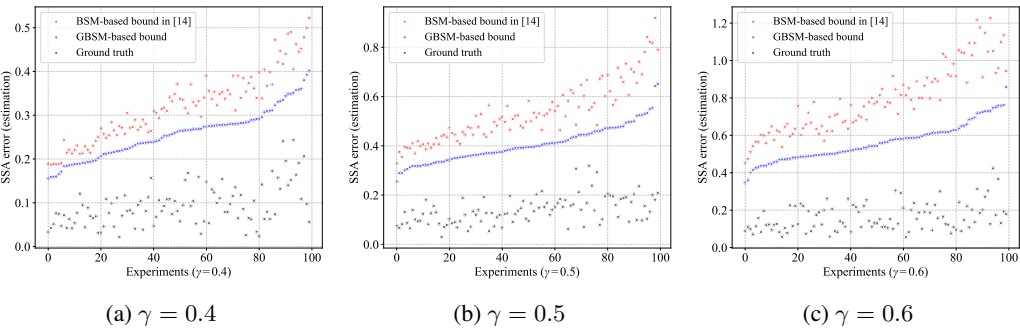

(a) $\gamma = 0.4$  (b) $\gamma = 0.5$  (c) $\gamma = 0.6$

Figure 9: Experiments on random Garnet MDPs (SSA with aggregation, $\gamma = 0.4$ to $0.6$).

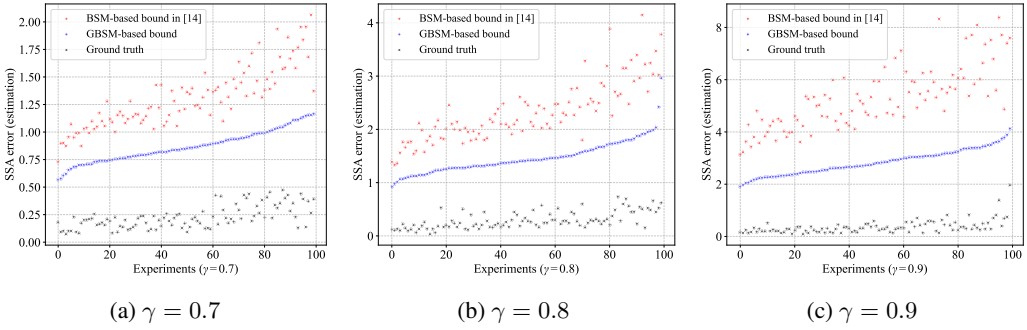

Figure 10: Experiments on random Garnet MDPs (SSA with aggregation, $\gamma = 0.7$ to $0.9$).

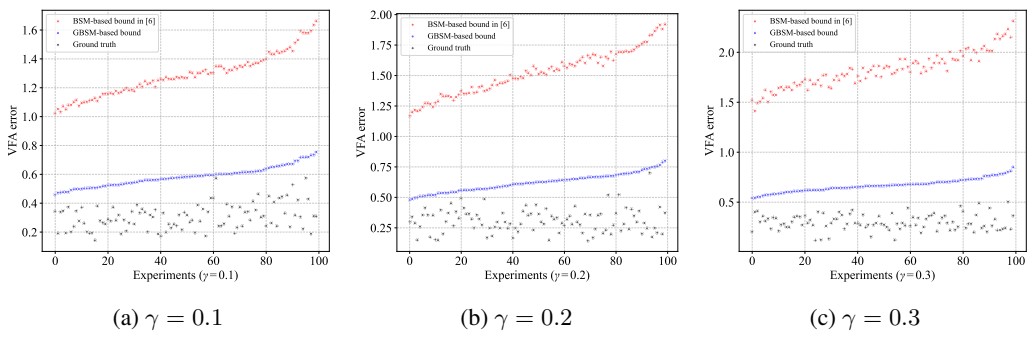

Figure 11: Experiments on random Garnet MDPs (VFA, $\gamma = 0.1$ to $0.3$).

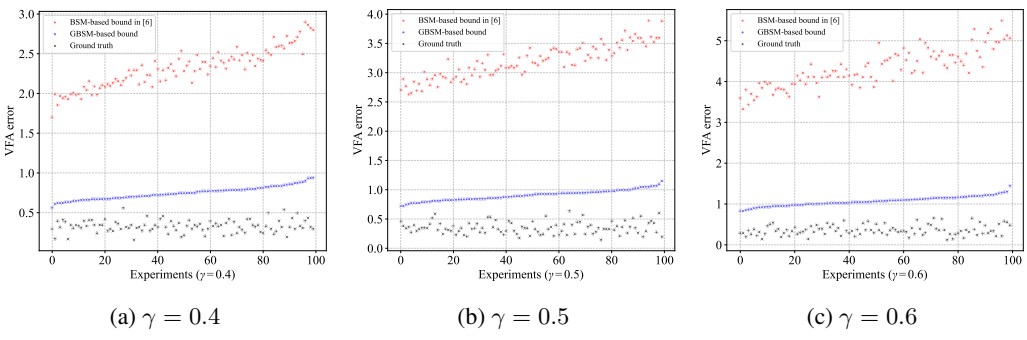

Figure 12: Experiments on random Garnet MDPs (VFA, $\gamma = 0.4$ to $0.6$).

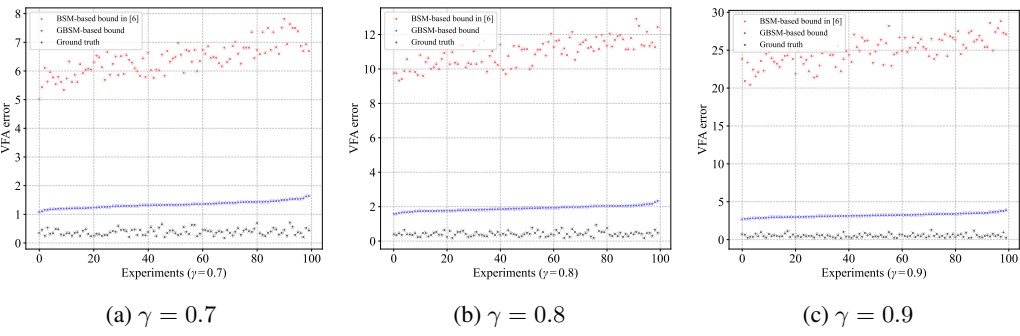

Figure 13: Experiments on random Garnet MDPs (VFA, $\gamma = 0.7$ to $0.9$).

