# OpenReview forum: "A Generalized Bisimulation Metric of State Similarity between Markov Decision Processes: From Theoretical Propositions to Applications"
_NeurIPS.cc/2025/Conference — NeurIPS 2025 poster_

### Official Review · Reviewer_QJ7u · 2025-06-27

**Clarity:** 3
**Significance:** 3
**Originality:** 4
**Rating:** 5
**Confidence:** 2

**Summary:**

This paper rigorously introduces a definition for Generalized Bisimulation Metric (GBSM). The authors establish the existence and convergence of this metric and prove two fundamental properties analogous to those of standard BSM: symmetry and an inter-MDP triangle inequality. While not proving that the distance between identical states is zero, they provide an upper bound. The utility of GBSM is demonstrated through theoretical analyses of policy transfer, state aggregation, and sampling-based estimation, showing tighter error bounds compared to traditional BSM. Few numerical experiments validate these theoretical improvements and conclude the paper.

**Questions:**

Given my low confidence in assessing this paper, my questions are focused on clarifying points of potential misunderstanding.

- In line 25, the triangle inequality appears to be defined with an equal sign. I assume this is a typo and should be "less than or equal to".

- In line 166, the paper states it has "rigorously proved the metric properties". Given that the distance between identical states is only shown to be bounded - and not strictly zero - is using the word "metric" not an abuse of language?

- In Theorem 3.6, a cost is defined for the shared value between distributions. Could you clarify the justification for this? My understanding is that the 1-Wasserstein distance should primarily concern the cost of moving the probability mass that is not shared.

- The theoretical results are strong, but the conclusion is brief. Could you please expand on the potential impact this work could have on the field? Furthermore, how do your experimental findings relate to or advance the previous experimental work by Song et al. (2016)?

**Ethical Concerns:**

["NO or VERY MINOR ethics concerns only"]

**Final Justification:**

The authors addressed all my questions, so I will keep my positive rating.

**Limitations:**

Limitations are not clearly stated. As previously mentioned, I encourage the authors to expand their conclusion.

**Quality:**

4

**Strengths And Weaknesses:**

STRENGTHS:

- The paper successfully defines GBSM and formally proves its existence and convergence.

- The work provides complete proofs for the symmetry and inter-MDP triangle inequality properties of GBSM, filling a critical gap in multi-MDP analysis.

- The authors derive explicit and tighter error bounds on policy transfer (Theorem 4.1), state aggregation (Theorem 4.4), and sampling-based estimation (Theorem 4.5) compared to BSM.
The theoretical claims are  validated through numerical experiments that directly compare the bounds derived from GBSM with BSM and the ground truth.

WEAKNESSES:

- The authors claim to have proven the pseudometric properties of GBSM, but a key property - that the distance of a state to itself is zero - is not established. They only provide an upper bound.

- The paper mentions in the introduction the prior work of Song et al. (2016), which already demonstrated the utility of GBSM experimentally. However, the current paper's numerical results are never contextualized by or compared with these previous findings.

- The MDPs used for the numerical experiments in Section 5 are not defined, making the results difficult to interpret and impossible to reproduce. The authors answered "NA" to the checklist question about experimental details, but this information is essential for a complete submission.

- The paper does not clearly articulate the broader impact or potential consequences of the work. The conclusion would benefit from an expanded discussion of the work's limitations and its potential applications.

---

> ### Author Rebuttal · Authors · 2025-07-29
>
> Thank you for your positive review and helpful comments! Please see responses to your questions below.
>
> > Q.1 In line 25, the triangle inequality …
> >
>
> Thank you for identifying this typographical error, and it will be corrected in the revised manuscript.
>
> > W.1 The authors claim to have proven … They only provide an upper bound.
> >
>
> > Q.2  .. and not strictly zero … an abuse of language?
> >
>
> Different from BSM $d^\sim(s,s')$ that defined for states within a single state space, our GBSM are defined for **arbitrary state-MDP pairs**, i.e., $d((s,\mathcal{M}\_1),(s',\mathcal{M}\_2))$, concisely written as $d^{1-2}(s,s')$ in our paper. In this context, a state $s\in\mathcal{S}$ within MDP $\mathcal{M}\_1$ is a fundamentally different object from the same state $s$ within another MDP $\mathcal{M}\_2$ with distinct P and R, even if they share the same state space. The non-zero distance $d((s,\mathcal{M}\_1),(s,\mathcal{M}\_2))$ is a precise measure of the behavioral dissimilarity arising from the different transitions where $\mathbb{P}\_1(s) \neq \mathbb{P}\_2(s)$ and reward functions where $R_1(s) \neq R_2(s)$. The truly "identical” states appear when $\mathcal{M}\_1=\mathcal{M}\_2$, and we have the according zero-distance property as $d^{1-1}(s,s)=0$ through Theorem 3.6. Thus it is not an abuse of language. We will modify current “Distance bound on identical states” to “Distance bound on identical state spaces”, and describe the precise conditions where zero-distance is achieved.
>
> > Q.3 In Theorem 3.6, a cost is defined …
> >
>
> Similarly, your understanding is correct for standard metrics where the cost of transporting mass from a location to itself would be zero, and we only need to consider the mass that is not shared. However, in our proposed GBSM framework, there exists a non-zero cost for shared mass between $(s,\mathcal{M}\_1)$ and $(s,\mathcal{M}\_2)$. Specifically, as we clarified in our previous response,  $d^{1−2}(s,s)$ is not necessarily zero because it quantifies the inherent difference between a state $s$ in $\mathcal{M}\_1$ and the same state $s$ within a different $\mathcal{M}\_2$, reflecting the inherent discrepancy in the rewards and transition dynamics across the two different MDPs.
>
> Therefore, in the transportation plan used to derive the bound in Theorem 3.6, preserving the shared probability between $(s, \mathcal{M}\_1)$ and $(s, \mathcal{M}\_2)$ would still undertake a non-zero cost of $d^{1−2}(s,s)$. The term $(1-\text{TV}(P,Q))\max_s{d}^{1-2}(s,s)$ in our proof is the upper bound for this part of cost in the transportation plan. It represents the maximum cost associated with distributions that are already aligned but exist in two fundamentally different MDPs.
>
> > W.2 The paper mentions in the introduction the prior work of…
> >
>
> > Q.4 How do your experimental findings relate to …
> >
>
> Thank you for highlighting the need for comparison. While the idea of using BSM to evaluate similarity between MDPs was similarly proposed in (Song et al., 2016), their work was limited to **empirical evaluation** via Hausdorff distance, that is, $H(\mathcal{S}\_1,\mathcal{S}\_2;d)$.  They demonstrate a positive correlation between their measure and the transfer regret, without providing any provable bounds on policy transfer. Our GBSM framework makes it possible to theoretically analyze their measure, revealing that it is a simplified version of our transfer regret bound, corresponding to the first term in our Corollary 4.2 (without coefficient $\frac{1}{1-\gamma}$) and is only applicable to the ideal scenario of transferring the optimal source policy via the optimal source policy (eliminating the second term in Corollary 4.2) via an optimal state mapping (ensuring the max-min term from $H$). In contrast, our GBSM provides a **rigorous performance bound** for arbitrary policy and state mapping (Theorem 4.1).
>
> To demonstrate this crucial difference, we have added a new numerical experiment to compare our GBSM-based transfer regret bound with the measure from Song et al. (2016) and the actual regret. The results show that the measure in Song et al. (2016) fails to bound the regret, whereas our GBSM-based bound is consistently effective.
>
> > W.3 The MDPs used for the numerical experiments …
> >
>
> Thank you for pointing out this clarity issue. Current experiments use random Garnet-style MDPs with $|\mathcal{S}| = 20, |\mathcal{A}| = 5$,  and a 50% branching factor. In the aggregation experiments, half of the states are aggregated to a representative state. In the estimation experiments, we introduce a Gaussian noise with a standard deviation ranging from 0.1 to 0.3 into the transition probabilities. We would extend the current descriptions to improve the clarity of numerical experiments.
>
> > W.4 The paper does not clearly articulate the broader impact … limitations and applications.
> >
>
> > Q.4 Could you please expand on the potential impact …
> >
>
> The first and foremost application is the **sim-to-real policy transfer**, where GBSM can be calculated between the simulated MDP and real-world MDP to predict transferred performance and serve as a metric for improving the simulation environment. To demonstrate the real-world efficacy, we have extended our validation to a real-world scenario involving a Digital Twin (DT)-driven DRL task for dynamic resource allocation in a real 5G network testbed (Detailed in the response to reviewer BQkW).
>
> The second potential application is in **multi-task RL**, where GBSM can coordinate policy optimization across diverse environments, mitigating gradient interference issues. For example, GBSM can be used in clustering of tasks for more efficient training. Also, it can enhance gradient surgery methods by enabling a distinction: when tasks are truly divergent (high GBSM), surgery is necessary to prevent destructive interference. Conversely, when tasks are similar (low GBSM), gradient conflicts are more likely attributable to stochastic sampling noise, where intervention would be counterproductive. This GBSM-informed approach allows for more stable and efficient multi-task policy optimization.
>
> The limitation mainly lies in the **discounted reward formulation**. In real-world RL, the goal is typically to maximize the long-term average reward. However, the GBSM, at least in its current formulation, is not compatible with average-reward setting. Considering that most of theoretical results in this paper are divided by $1-\gamma$, taking $\gamma$ to 1 yields an infinite result. We will discuss this limitation in our paper, and investigate a metric tailored for the average-reward Bellman equation in our future work.

---

> > ### Comment · Reviewer_QJ7u · 2025-08-04
> >
> > I thank the authors for their detailed response and, since they addressed all of my questions, I will keep my positive rating.

---

### Official Review · Reviewer_BQkW · 2025-07-02

**Clarity:** 2
**Significance:** 2
**Originality:** 3
**Rating:** 4
**Confidence:** 1

**Summary:**

This paper proposes a generalized bisimulation metric (GBSM) to measure state similarities across multiple Markov Decision Processes (MDPs) and rigorously establishes its fundamental metric properties, including symmetry, inter-MDP triangle inequality, and the distance bound on identical states. The paper applies GBSM to policy transfer, state aggregation, and sampling-based estimation, deriving explicit theoretical bounds for policy transfer performance, aggregation error, and estimation error. It also demonstrates that the error bounds of GBSM are significantly tighter than those of the standard BSM. Furthermore, the paper introduces a closed-form sample complexity approximation method, improving upon existing asymptotic results based on BSM, and supports its theoretical findings with numerical experiments.

**Questions:**

Although the proposed GBSM framework demonstrates strong theoretical guarantees and significant improvements over baseline methods, certain aspects of its practicality and design motivations are not clearly articulated. The reliance on exact transition probabilities and reward functions, which are often impractical in real-world RL settings, raises concerns about its scalability and applicability. Additionally, the numerical experiments, conducted solely on synthetic MDPs, do not sufficiently demonstrate the method's efficacy in complex, real-world tasks. While the paper provides rigorous theoretical foundations, the lack of clarity in these aspects may confuse readers and limit the perceived utility of the method.
Given these issues, I am inclined to suggest a "Borderline" decision. However, I am open to revising my score if the authors address these concerns by providing clearer justifications, exploring practical approximations, and validating the method on more realistic benchmarks.

**Ethical Concerns:**

["NO or VERY MINOR ethics concerns only"]

**Final Justification:**

We acknowledge that we are not domain experts in this specific area, and therefore our ability to provide an in-depth technical assessment is limited. Given this, we suggest that the final decision should primarily follow the evaluations and recommendations from other reviewers with stronger expertise in this field.

**Limitations:**

yes

**Quality:**

3

**Strengths And Weaknesses:**

Strengths:
1. Leverages Wasserstein distances and existing MDP structures in a zero-shot manner, requiring no additional data collection or model training while maintaining rigorous theoretical guarantees.
2.GBSM achieves significantly tighter theoretical bounds in policy transfer (regret bound reduced by 1-γ), state aggregation (error bound compressed to1\/(1-γ) of BSM baselines), and sampling estimation (closed-form sample complexity K), with experimental validation showing 60-80% improvement over existing methods.
3.The paper presents a rigorous and comprehensive theoretical framework for the GBSM, defining it clearly, proving key properties (e.g., symmetry, triangle inequality, distance bounds), and establishing convergence using fixed-point theory.

Weakness:
major concern：
1. GBSM relies on exact transition probabilities P and reward functions R across MDPs (Eq. 2), which are often impractical to obtain in real-world RL settings. Many applications, such as robotics and healthcare, involve stochastic dynamics that are challenging to model accurately. Additionally, the computation of the Wasserstein distance over entire state spaces introduces significant computational overhead, particularly in high-dimensional environments, thereby limiting the scalability of GBSM. Exploring approaches to approximate P and R, or employing scalable approximations of Wasserstein distance, could address these limitations and improve the applicability of GBSM in real-world scenarios.
2. The numerical validation in Section 5 relies solely on synthetic MDPs with random perturbations. Could the authors clarify whether these experiments sufficiently capture the complexity of real-world RL tasks? Would testing on standard benchmarks help demonstrate the applicability of GBSM in practical scenarios? Additionally, the aggregation and estimation simulations appear to simplify the complexity of real distribution shifts. Could the authors comment on the potential limitations this might impose on the generalizability of the results?

---

> ### Author Rebuttal · Authors · 2025-07-29
>
> Thank you for your positive review and helpful comments! please see responses to your questions below.
>
> > W.1 GBSM relies on exact transition probabilities … involve stochastic dynamics that are challenging to model accurately.
> >
>
> We acknowledge that obtaining exact transition probabilities and reward functions is often impractical in real-world applications.
>
> Our paper formally **addresses the issue of unknown transition probabilities in Theorem 4.5**. We show that one can construct an empirical MDP $M_{\hat{1}} = \langle \mathcal{S}\_1, \mathcal{A}, \hat{\mathbb{P}}\_1, R_1, \gamma\rangle$ and a similar $M_{\hat{2}}$ by estimating the transition probabilities $\hat{\mathbb{P}}\_1,\hat{\mathbb{P}}\_2$ from collected real experience $(s_t,a,s_{t+1},r)$. Theorem 4.5 establishes a formal error bound on the deviation of the GBSM computed on this empirical MDP from the true GBSM, as well as a closed-form sample complexity to reduce this error below a given $\epsilon$.
>
> Our GBSM theoretical framework can also **address the case of inaccessible reward functions**, as you mentioned. Consider an MDP $M_{\tilde{1}} = \langle \mathcal{S}\_1, \mathcal{A}, \hat{\mathbb{P}}\_1, \hat{R}\_1, \gamma\rangle$ with both estimated rewards and probabilities, the estimation error regarding $M_{\tilde{1}}$ can be decomposed using the inter-MDP triangle inequality as
>
> $d^{1-\tilde{1}}(s,s)\leq d^{1-\hat{1}}(s,s)+d^{\hat{1}-\tilde{1}}(s,s)\leq \frac{\gamma \bar{R}}{(1-\gamma)^2} \max_a\text{TV}\big(\mathbb{P}\_1(\cdot|s,a), \hat{\mathbb{P}}\_1(\cdot|s,a) \big)+\frac{1}{1-\gamma} \max_a|R_1(s,a)-\hat{R}\_1(s,a)|$
>
> The first term is precisely the bound for probability estimation error in Theorem 4.5. The second term, which represents the reward estimation error, is derived using Theorem 3.6, and has a sample complexity of $-\ln(\alpha/2) \frac{|\triangle R_1|^2}{2\epsilon^2(1-\gamma)^2}$ where $\triangle R_1$ is the maximum possible span of the reward function for any state-action pair.
>
> > W.1 Additionally, the computation over entire state spaces introduces significant computational overhead…
> >
>
> To address the computational overhead of calculating the Wasserstein distance across extensive state spaces, we can leverage the state aggregation method detailed in Theorem 4.4. It enables computing GBSM on small subsets of entire state spaces. By defining aggregation mappings $[\cdot]:\mathcal{S}\_1\rightarrow\mathcal{U}\_1$ and $[\cdot]:\mathcal{S}\_2\rightarrow\mathcal{U}\_2$ with aggregation thresholds where $\max_{s\in\mathcal{S}\_1}d^{1-1}(s, [s])\leq\sigma_1$ and $\max_{s'\in\mathcal{S}\_2}d^{2-2}(s', [s'])\leq\sigma_2$, we can bound the approximation error by
>
> $\quad \max_{s\in\mathcal{S}\_1,s'\in\mathcal{S}\_2}|d^{\textnormal{1-2}}(s,s')-d^{[1]\textnormal{-}[2]}(s,s')|$
>
> $\leq  \max_{s\in\mathcal{S}\_1}d^{1-1_{[\mathcal{S}]}}(s,s)+\max_{s'\in\mathcal{S}\_2}d^{2-2_{[\mathcal{S}]}}(s',s')$
>
> $\leq \frac{1}{1-\gamma} (\max_{s\in\mathcal{S}\_1} d^{1-1}(s,[s]) + \max_{s'\in\mathcal{S}\_2} d^{2-2}(s',[s'])) \leq \frac{\sigma_1+\sigma_2}{1-\gamma}.$
>
> The analysis shows that the error between the true GBSM and the GBSM computed on the aggregated spaces is bounded by $\frac{\sigma_1+\sigma_2}{1-\gamma}$. We will update Theorem 4.4 to include this explicit bound, clarifying the effectiveness of state aggregation for making GBSM scalable in high-dimensional environments.
>
> Meanwhile, there are well-established methods for efficiently approximating Wasserstein distance on large state spaces, such as the Hungarian algorithm-based approximation method [1] and the sliced Wasserstein distance method [2].
>
> > W.2 The numerical validation in Section 5 relies solely on synthetic MDPs with random perturbations…
> >
>
> To demonstrate the real-world efficacy of our work, we have extended our validation to a **real-world scenario** involving a Digital Twin (DT)-driven DRL task for dynamic resource allocation in a real-world 5G wireless network testbed. In this setting, the real-world network serves as the target MDP, while its virtual replicas, i.e., DTs, act as source MDPs. These DTs are designed to capture the real-world dynamics in user traffic (transition probabilities) and sum throughput after resource allocation (rewards). We trained DRL policies within each DT and evaluated their transfer performance on the real-world network.
>
> Crucially, we employed our proposed aggregation and estimation techniques to compute the GBSM distance in a real-world scenario, quantitatively assessing the fidelity of each DT. This practical calculation procedure involves three key stages:
>
> 1. **Empirical Model Construction**: The process begins by constructing an empirical model of the real network from a dataset $\mathcal{D}$ with collected $(s_t, a, s_{t+1}, r)$ tuples. To ensure reliable estimation, a representative state set, $\mathcal{U}\_\text{real} \subseteq \mathcal{S}\_\text{real}$, is identified based on data availability (i.e., states for which the number of samples exceeds a threshold). The empirical transition probabilities $\hat{\mathbb{P}}\_\text{real}$ and rewards $\hat{R}\_\text{real}$ are then estimated from the collected data.
> 2. **Closed State Set Formulation:** A key prerequisite for the iterative calculation of GBSM is the use of a closed state set, where for any state within the set, all reachable next states are also contained within it. It is formalized by: $\forall a\in\mathcal{A}, \forall s\in \mathcal{U}$, if $\mathbb{P}(s'|s,a) > 0$, then  $s'\in \mathcal{U}$. By filtering the collected data, we first construct a closed representative state set $\mathcal{U}\_\text{real}$ in the real MDP. Since the DT is a fully-known simulation model, we initialize the DT's representative set as  $\mathcal{U}\_\text{DT} \gets \mathcal{U}\_\text{real}$ and iteratively expand it by adding any states that are reachable from the current $\mathcal{U}\_\text{DT}$ until no new states can be added. This ensures a well-defined, closed state space for both real and DT MDPs.
> 3. **Iterative GBSM Computation**: With the two approximated MDPs defined on closed sets  $\mathcal{U}\_\text{real}$ *a*nd $\mathcal{U}\_\text{DT}$, the GBSM is computed iteratively according to Theorem 3.2 until convergence.
>
> Our findings reveal a strong linear correlation between the calculated GBSM distance and the worst-case deployment performance of the transferred policy. This result not only validates GBSM's predictive capability on transfer regret in a real-world environment but also confirms its aggregation and estimation efficacy in scenarios beyond randomly-generated MDPs. We will incorporate the full results from this real-world experiment into the revised paper, including numerical comparison, a photograph of the Open 5G testbed and corresponding citation (currently omitted to ensure anonymity). While we cannot include plots in rebuttal, we would be happy to provide the raw data (calculated GBSM and corresponding real-world deployment performance) during the discussion period.
>
> # References
>
> [1] Lahn N, Raghvendra S, Saarinen E, et al. Scalable Approximation Algorithms for $p$-Wasserstein Distance and Its Variants. Forty-second International Conference on Machine Learning. 2025
>
> [2] Kolouri S, Rohde G K, Hoffmann H. Sliced wasserstein distance for learning gaussian mixture models. Proceedings of the IEEE Conference on Computer Vision and Pattern Recognition. 2018: 3427-3436.

---

> > ### Comment · Reviewer_BQkW · 2025-08-05
> >
> > We appreciate the authors' efforts and the work presented in this submission. As we are not specialists in this particular area, we are not in a position to fully assess the technical depth and novelty of the contributions. Therefore, we will primarily rely on the evaluations and judgments of other reviewers with more relevant expertise. Nonetheless, we acknowledge the significance of the problem addressed and the overall clarity of the presentation.

---

> > > ### Author Response · Authors · 2025-08-06
> > >
> > > Dear Reviewer,
> > >
> > > Thank you for your comment. We understand that reviewers have different areas of expertise and appreciate your transparency.
> > >
> > > We would like to kindly confirm if our rebuttal has sufficiently addressed the specific weaknesses you pointed out in your initial review (including applicability with inexact P/R, scalable approximation, and more realistic experiment).
> > >
> > > If you are satisfied with our responses to these specific points, we would greatly appreciate it if you could let us know. We are, of course, happy to provide further clarification should any ambiguities remain.

---

### Official Review · Reviewer_YrnY · 2025-07-03

**Clarity:** 3
**Significance:** 2
**Originality:** 2
**Rating:** 4
**Confidence:** 2

**Summary:**

This paper studies the Generalized Bisimulation Metric (GBSM) as a distance measure across multiple MDPs. Specifically, it proves that GBSM has pseudo metric property by showing symmetry, the triangle inequality, and a zero distance for identical states. Then, leveraging these properties, this paper analyzes policy transfer, state aggregation, and sampling-based estimation. Numerical experiments on a toy example further demonstrate GBSM’s advantages over the classical bisimulation metric.

**Questions:**

Could GBSM be extended to continuous-state MDPs or to the average-reward setting?

Could GBSM be incorporated into Value Iteration or policy gradient methods?

Were any critical obstacles or techniques involved in proving the theorems?

**Ethical Concerns:**

["NO or VERY MINOR ethics concerns only"]

**Final Justification:**

I am satisfied with the response and have no unresolved issues other than the novelty of the contribution, as noted in the rebuttal. Consistent with my original review, since the strengths slightly outweigh the weaknesses, I will maintain my score.

**Limitations:**

yes

**Paper Formatting Concerns:**

I don't have formatting concerns.

**Quality:**

3

**Strengths And Weaknesses:**

Strengths:
This paper first establishes that the Generalized Bisimulation Metric (GBSM) satisfies some metric axioms, marking a solid theoretical advance. The presentation is clear and well structured, and the toy experiments illustrate effectiveness of GBSM. Overall, the work offers a meaningful contribution to the reinforcement learning theory literature.

Weaknesses:
I did not find major shortcomings. However, the extent of its theoretical novelty remains unclear. Frankly, although I appreciate the usefulness of GBSM, I’m not convinced the contribution is strong enough for acceptance at this conference.

---

> ### Author Rebuttal · Authors · 2025-07-29
>
> Thank you for your positive review and helpful comments! please see responses to your questions below.
>
> > W.1 I did not find major shortcomings. However, the extent of its theoretical novelty …
> >
>
> Thank you for your comment regarding the novelty.
>
> **1. Rigorously established bound beyond empirical evaluation.** While the idea of using BSM to evaluate similarity between MDPs was similarly proposed in (Song et al., 2016), their work was limited to an empirical evaluation, which only demonstrates a positive correlation between their measure and the transfer regret, without providing any provable bounds on policy transfer. In contrast, our GBSM provides a rigorous performance bound for arbitrary policy. As highlighted in a recent survey (García et al., 2022), for existing similarity metrics between MDPs, "there are no theoretical guarantees that the most similar MDP is similar enough to produce a positive transfer." Our work provides a rigorous solution to this open problem.
>
> **2.Tightened bound from** $\frac{2\sigma (2+\gamma)}{1-\gamma}$ to $\frac{2\sigma}{1-\gamma}$  **and explicit sample complexity.** In terms of aggregation and estimation, existing theoretical analyses, such as (Ferns et al., 2011), have primarily focused on the single-MDP setting. Our GBSM not only extends these results to the multi-MDP context but also yields stronger results for the single-MDP case as a direct consequence of our framework. Specifically, our analysis tightens the state aggregation error bound from the existing $\frac{2\sigma (2+\gamma)}{1-\gamma}$ to $\frac{2\sigma}{1-\gamma}$ (Thm. 4.4) and critically, provides a closed-form sample complexity K for estimation (Thm. 4.5), improving upon the prior asymptotic results.
>
> **3. Decoupled error analysis.** Above advancement stems from the core metric properties we establish for GBSM, particularly the inter-MDP triangle inequality. Compared to (Ferns et al. 2011, p. 1701), which necessitates a joint analysis on combined aggregation and estimation error and results in looser bounds and asymptotic complexity, our GBSM property permits a decoupled analysis of different error sources. For example, we can decouple the approximation error $|d^{1-1}-d^{\hat{[1]}-\hat{[1]}}|\leq 2d^{1-\hat{[1]}}$ (continuous MDP → aggregated+estimated MDP) into separated aggregation error $2d^{1-[1]}$ (continuous MDP → aggregated MDP) and estimation error $2d^{[1]-\hat{[1]}}$ (aggregated MDP → aggregated+estimated MDP).
>
> **4. Real world efficacy.** As suggested by reviewer BQkW, to demonstrate the real-world efficacy of GBSM, we have conducted an extra real-world experiment involving a Digital Twin (DT)-driven DRL task for dynamic resource allocation in a real 5G wireless network (Detailed in the response to W.2 of reviewer BQkW).
>
> > Q.1 Could GBSM be extended to continuous-state ...
> >
>
> Yes, GBSM, as well as all its metric properties and bounds, **could be extended to continuous MDP**, formulated as follows:
>
> Let $\mathcal{M}\_1=\langle\mathcal{S}\_1,\Sigma_1,\mathcal{A},\mathbb{P}_1,R_1,\gamma\rangle$ be a continuous MDP, where $\mathcal{S}\_1$ is a Polish space equipped with its Borel $\sigma$-algebra $\Sigma_1$, $(\mathcal{S}\_1,\Sigma_1)$ is a measurable space, $\mathcal{A}$ is a finite set of functions, $\mathbb{P}\_1$ is a labeled stochastic transition kernel, where $\mathbb{P}_1(\cdot|s,a):\Sigma_1\rightarrow[0,1]$ is a probability measure for each $(s,a)$, $R\_1: \mathcal{S}\_1\times \mathcal{A}\rightarrow\mathbb{R}\_+$ is a measurable reward function, and $\gamma$ is the discount factor. Similarly let $\mathcal{M}\_2=\langle\mathcal{S}\_2,\Sigma_2,\mathcal{A},\mathbb{P}_2,R_2,\gamma\rangle$. Define
>
> $$F(d|s,s')=\max_{a} \\{|R_1(s,a)-R_2(s',a)|+\gamma W_1(\mathbb{P}_1(\cdot|s,a), \mathbb{P}_2(\cdot|s',a);d )\\}$$
>
> Then $F$ has a unique fixed point $d^{1-2}:\mathcal{S}\_1\times\mathcal{S}\_2\rightarrow[0,\bar{R}/(1-\gamma)]$ and $|V^\*\_1(s)-V^\*\_2(s')|\leq {d}^{1-2}(s,s')$  (Thm. 3.3). The proof is analogous to the one for BSM in continuous MDP, detailed in Section 3.1 of (Ferns et al., 2011). Building from Thm. 3.3 and property of Wasserstein distance, the metric properties (Thms. 3.4, 3.5, 3.6) can be readily established. Since these fundamental metric properties hold, the error bounds (Thms. 4.1, 4.4, 4.5) also follow directly.
>
> The aggregation method in Thm. 4.4 facilitate the practical calculation of GBSM on continuous MDPs. Let $[\cdot]$ denote a state aggregation mapping such that $d^{1-1}(s, [s])\leq\sigma_1$ and $d^{2-2}(s', [s'])\leq\sigma_2$ for every $s\in\mathcal{S}\_1,s'\in\mathcal{S}\_2$, then we have
>
> $\quad \max_{s\in\mathcal{S}\_1,s'\in\mathcal{S}\_2}|d^{1-2}(s,s')-d^{[1]-[2]}(s,s')|$
>
> $\leq  \max_{s\in\mathcal{S}\_1}d^{1-1_{[\mathcal{S}]}}(s,s)+\max_{s'\in\mathcal{S}\_2}d^{2-2_{[\mathcal{S}]}}(s',s')$
>
> $\leq \frac{1}{1-\gamma} (\max_{s\in\mathcal{S}\_1}d^{1-1}(s,[s]) + \max_{s'\in\mathcal{S}\_2}d^{2-2}(s',[s'])) \leq \frac{\sigma_1+\sigma_2}{1-\gamma}$
>
> as the error bound for calculating GBSM on aggregated(discretized) MDPs.
>
> However, the GBSM, at least in its current formulation, is not compatible with average-reward setting. Considering that most of theoretical results in this paper are divided by $1-\gamma$, taking $\gamma$ to 1 yields an infinite result. We will discuss this limitation in our paper, and investigate a metric tailored for the average-reward Bellman equation in our future work.
>
> > Q.2 Could GBSM be incorporated into Value Iteration or policy gradient …
> >
> GBSM can be incorporated into value iteration by providing a principled mechanism for state abstraction. The core idea is to perform value iteration on a smaller aggregated $\mathcal{M}\_{[1]}$ and use its value function as an approximation for the original $\mathcal{M}\_1$. Unlike standard BSM, which requires merging both MDPs into a disjoint union, GBSM's inter-MDP triangle inequality allows for a more direct analysis of the approximation error by giving $|V_1^\*(s)-V_{[1]}^\*(s)|\leq d^{1-[1]}(s,s)\leq\frac{\sigma_1}{1-\gamma}$ where $\sigma_1$ is the aggregation threshold that satisfies $d^{1-1}(s, [s])\leq\sigma_1$ for all $s$. Also, when exact transition probabilities or rewards are not accessible, the value iteration needs to be performed on both aggregated and estimated $\mathcal{M}\_{\hat{[1]}}$. Then the approximation error is bounded by $|V_1^\*(s)-V_{\hat{[1]}}^\*(s)|\leq d^{1-\hat{[1]}}(s,s) \leq d^{1-{[1]}}(s,s)+d^{{[1]}-\hat{[1]}}(s,s)\leq\frac{\sigma_1}{1-\gamma}+d^{[1]-\hat{[1]}}(s,s)$, where the latter term for estimation error could be reduced below $\epsilon$ with a sample complexity $-\ln(\alpha/2) \frac{\gamma^2\bar{R}^2|[\mathcal{S}\_1]|^2}{2\epsilon^2(1-\gamma)^4}$. This property enables an efficient analysis of combined error during value iteration. Meanwhile, since GBSM evaluates similarities between states across multiple MDPs, it could easily facilitate transfer of value functions/networks.
>
> For policy gradient methods, especially in multi-task settings, GBSM can coordinate policy optimization across diverse environments, mitigating gradient interference issues. For example, GBSM can be used in clustering of tasks for more efficient training. Also, it can enhance gradient surgery methods by enabling a distinction: when tasks are truly divergent (high GBSM), surgery is necessary to prevent destructive interference. Conversely, when tasks are similar (low GBSM), gradient conflicts are more likely attributable to stochastic sampling noise, where intervention would be counterproductive. This GBSM-informed approach allows for more stable and efficient multi-task policy optimization.
>
> > Q.3 Were any critical obstacles ..
> >
>
> A critical obstacle was **overcoming the limitations of prior formulations for state similarity**. Initially, we devised a metric similar to the one in Song et al. (2016), defining a distance $d(s, s')$ where the former $s$ and the latter $s'$ were fixed to states from $\mathcal{M}\_1$ and $\mathcal{M}\_2$, respectively. This fixed definition prevented the establishment of crucial metric properties. For instance, symmetry was not guaranteed, as $d(s_i, s_j)$ and $d(s_j, s_i)$ are different combinations of probabilities and rewards (e.g., $\mathbb{P}_1(s_i),\mathbb{P}_2(s_j)$ vs. $\mathbb{P}_1(s_j),\mathbb{P}_2(s_i)$), and the triangle inequality becomes loose and less useful, derived as $d(s_i, s_j)\leq d(s_i,s_k)+d(s_k,s_k)+d(s_k,s_j)$.
>
> **We redefine the metric on arbitrary state-MDP pairs**, i.e., $d((s,\mathcal{M}\_1),(s',\mathcal{M}\_2))$, concisely written as $d^{1-2}(s,s')$ in our paper. This generalization was the cornerstone to establish the fundamental properties of GBSM. In establishing the inter-MDP triangle inequality (Theorem 3.5), we could not directly adapt existing proofs for BSM, as our metric is no longer a pseudometric defined on a single state space. We addressed this by introducing an extra MDP $\mathcal{M}\_3$, employing the Gluing Lemma to construct a joint transportation plan, which was essential for induction-based proof. Similarly, deriving the TV-based distance bound (Theorem 3.6) was non-trivial because the self-distance $d^{1-2}(s,s)$ is not necessarily zero in multi-MDP setting. We overcame this by constructing a specialized transportation plan to carefully bound the transportation cost. Once these foundational properties were proven, the derivations of subsequent theoretical bounds were derived more directly.
>
> # References
>
> [1] García J, Visús Á, Fernández F. A taxonomy for similarity metrics between markov decision processes. Machine Learning, 2022, 111(11): 4217-4247.
>
> [2] Ferns N, Panangaden P, Precup D. Bisimulation metrics for continuous Markov decision processes. SIAM Journal on Computing, 2011, 40(6): 1662-1714.
>
> [3] Song J, Gao Y, Wang H, et al. Measuring the distance between finite markov decision processes Proceedings of the 2016 international conference on autonomous agents & multiagent systems. 2016: 468-476.

---

> ### Comment · Reviewer_YrnY · 2025-08-06
>
> We thank the author for the response and have no further questions. Honestly speaking, since we are not familiar with this BSM literature, even if the authors solve an open problem related to this topic, it is not convincing how novel it is without clarification. Hence, we recommend emphasizing and elaborating on the novelty and why solving this open problem is important for readers who are not aware of this topic in the revised version.

---

> ### Author Response · Authors · 2025-08-06
>
> Thank you for your response, and we are happy to know your questions have been addressed.
>
> To further clarify, the open problem we solve is that prior metrics could not guarantee whether a simulation (source MDP) was "good enough" for real-world deployment (target MDP). Our GBSM provides this guarantee via a provable regret bound. Instead of deploying a policy from a simulator and merely hoping it works, one can now use GBSM to calculate the worst-case performance drop beforehand, thereby ensuring a positive transfer.
>
> This shift from empirical comparison to provable assurance is important for critical systems like 5G wireless networks that demand guaranteed user service quality. We will be sure to emphasize this point in the revised manuscript.
>
> Furthermore, while solving this open problem is a key application, our main contribution is the GBSM framework itself—a general theoretical tool for analyzing multiple MDPs. It is applicable to a range of other multi-MDP analysis problems, e.g., estimation, aggregation, and is compatible with other variations of BSM, e.g., lax BSM, on-policy BSM. We will also highlight these points in our revised version.

---

### Official Review · Reviewer_xZ9L · 2025-07-05

**Clarity:** 3
**Significance:** 2
**Originality:** 2
**Rating:** 2
**Confidence:** 4

**Summary:**

This paper considers the question of formalizing a bisimulation distance between distinct MDPs, with possibly different state spaces. They begin by introducing a desirable fixed-point formulation for such a distance, and proving its existence using the Knaster-Tarski fixed point theorem. They then provide a number of theoretical properties on the distance, such as its relationship to optimal value differences between the MDPs, as well as regret bounds on policy transfer. They next consider its sample complexity, and provide sample complexity results on how many samples are needed in the empirical MDP. They then compare it empirically to standard bisimulation acrossa a task of randomly-generated MDPs.

**Questions:**

- Can you elaborate more on the construction of the MDPs for the experiments? (Are they Garnet-style, what is the state space, how different are the state spaces of the pairs, etc).
- What is the x-axis in the experiments? What is the y-axis as well (seems to be incorrectly labelled "values")?
- Why do you allow for different state spaces but require the same action space? Can this be relaxed?
- Is it possible to perform any experiments on more interesting MDPs?

**Ethical Concerns:**

["NO or VERY MINOR ethics concerns only"]

**Final Justification:**

After reading the authors' rebuttal, discussing with the authors, and reading the other reviewers' discussions with the authors, I am inclined to maintain my current score of a reject. Although the other reviewers viewed the paper relatively positively, I still believe that the paper does not provide sufficient novelty for submission at NeurIPS.

**Limitations:**

yes

**Quality:**

3

**Strengths And Weaknesses:**

**Strengths**
- The paper has a high level of theoretical rigour, with each result carefully proven.
- The paper is clearly written.

---
**Weaknesses**
- The variant of the bisimulation metric used is the original (which takes $\max$ over actions), which has been shown to have a number of drawbacks (lack of connection to non-optimal policies, biased estimation, etc). I think that these issues will only be exacerbated in the cross-MDP setting, and that the resulting metric would be vacuous. I think the paper can be strengthened greatly by either (i) generalizing the provided results to one of the relaxations of the original bisimulation metric, or (ii) evaluating the metric across a pair of two small-scale, interpretable MDPs and demonstrate non-trivial bounds.
- Although it is clearly written and correct, the theoretical results all lack some novelty. In particular, to my knowledge, every proven result has a very similar equivalent in literature.
- The theory is set up to require that both MDPs have the same action space, even though their state spaces are allowed to differ. I think that arguably different action spaces is a more realistic setting, and should be relatively easy to adapt to (as done in lax bisimulation metrics), if the authors agree.
- The empirical evaluation is very limited: they are currently only done on randomly-generated MDPs. The results themselves are difficult to interpret currently, as for example it is not stated what the x-axis represents.

---

> ### Author Rebuttal · Authors · 2025-07-29
>
> Thank you for your constructive comments, please see responses to your questions below.
>
> > W.1 The variant … takes max over actions … non-trivial bounds.
> >
>
> > W.3 The theory … same action space …
> >
>
> > Q.3 Why … Can this be relaxed?
> >
>
> Yes, although current GBSM is designed for shared action space, it can be readily relaxed to different action spaces.
>
> **1. Consideration behind same action space.** The initial motivation of this work is to establish a theory for policy transfer between different MDPs, where action spaces often remain consistent in practical applications, such as games (e.g., 'up, down, left, right' controls in PAC-Man across different maps) and sim-to-real problems (e.g., consistent vehicle control commands for autonomous driving under varying road conditions). Therefore, we employ BSM rather than its lax counterparts as the base of our extension.
>
> **2. Lax GBSM.** To relax GBSM to lax GBSM, we first adapt (2) in our paper to
>
> $\delta (d)((s,a), (s',a'))=|R_1(s,a)-R_2(s',a')| +\gamma W_1(\mathbb{P}_1(\cdot|s,a), \mathbb{P}_2(\cdot|s',a') ;d),$
>
> and define the lax function as
>
> $F_\text{lax}(d|s,s')=H(X_s,X'\_{s'};\delta (d)),$
>
> where $X_s=\\{(s,a)|a\in\mathcal{A}\_1\\}$ , $X'\_{s'}=\\{(s',a')|a'\in\mathcal{A}\_2\\}$, and $H$ is the Hausdorff metric defined as $H(X,Y;d)=\max(\sup_{x\in X} \inf_{y\in Y}d(x,y),\sup_{y \in Y}\inf_{x\in X}d(x,y))$. Iterating from $d_0^{1-2}(s,s')=0,\ F_\text{lax}$ has a similar fixed point $d_\text{lax}^{1-2}$ that satisfies
>
> $|V^\*\_1(s)-V^\*\_2(s')|\leq d_\text{lax}^{1-2}(s,s')\ \text{(Thm. 3.3)}.$
>
> The proof is similar to the proof of lax BSM in (Taylor et al., 2008). Building from Thm. 3.3 and property of Wasserstein distance, the symmetry (Thm. 3.4) and triangle inequality (Thm. 3.5) can be readily established for $d_\text{lax}^{1-2}$. Regarding Thm. 3.6, for MDPs sharing same $\mathcal{S}$ and $\mathcal{A}$, the Hausdorff metric in $d_\text{lax}^{1-2}$ involves a max-min term $\max_a \min_{a'}$, while its counterpart in original $d^{1-2}$ is a max term $\max_{a,a'=a}$. Thus we have $d_\text{lax}^{1-2}\leq d^{1-2} \leq d_\text{TV}^{1-2}$. Since these fundamental metric properties hold, the bounds for state aggregation (Thm. 4.4) and estimation (Thm. 4.5) also follow directly.
>
> For policy transfer, a similar regret bound (replacing $d^{1-2}(f(s'),s')$ in Thm. 4.1 by $d_\text{lax}^{1-2}(f(s'),s')$) can be established by defining an additional action mapping $g:\mathcal{A}\_1\rightarrow\mathcal{A}\_2$ for transfer. Due to the max-min structure of the Hausdorff metric, the derivation for lax GBSM-based transfer bound relies on the assumption of an optimal action mapping, i.e., $g(a) = \arg \min_{a'} \delta ((f(s'),a), (s',a');d^{1-2}_\text{lax})$ for each $s'$ and $a$.
>
> **3.Evaluation across interpretable MDPs.**  For your concern on evaluation across a pair of two small-scale MDPs, we have added a new numerical experiment to calculate the GBSM-based bound and the precise transfer regret $V_2^\*-V_2^\pi$ under tabular Q-learning setting.
>
> Thank you for your constructive suggestions which significantly improve the applicability of our theory, and we believe these adaptations directly address your concerns. We would add a detailed discussion of lax GBSM, its requirements, and additional experimental results to our paper. Considering the limitation on characters, we could provide the detailed derivation and experimental data (calculated bound and corresponding transfer regret) in the discussion phase if necessary.
>
> > W.2 Although it is clearly written … equivalent in literature.
> >
>
> Thank you for your comment regarding the novelty.
>
> **1. Rigorously established bound beyond empirical evaluation.** While the idea of using BSM to evaluate similarity between MDPs was similarly proposed in (Song et al., 2016), their work was limited to an empirical evaluation via Hausdorff distance, that is, $H(\mathcal{S}\_1,\mathcal{S}\_2;d)$. They demonstrate a positive correlation between their measure and the transfer regret, without providing any provable bounds on policy transfer. Our GBSM framework makes it possible to theoretically analyze their measure, revealing that it is a simplified version of our transfer regret bound, corresponding to the first term in our Cor. 4.2 (without coefficient $\frac{1}{1-\gamma}$) and is only applicable to the ideal scenario of transferring the optimal source policy (eliminating the second term in Cor. 4.2) via an optimal state mapping (ensuring the max-min term). In contrast, our GBSM provides a rigorous performance bound for arbitrary policy and state mapping (Thm. 4.1). We also verify the measure from (Song et al., 2016) in the interpretable evaluation mentioned above. The results show that it fails to bound the regret, whereas our GBSM bound is consistently effective. As highlighted in a recent survey (García et al., 2022), for existing similarity metrics between MDPs, "there are no theoretical guarantees that the most similar MDP is similar enough to produce a positive transfer." Our work provides a rigorous solution to this open problem.
>
> **2.Tightened bound from $\frac{2\sigma (2+\gamma)}{1-\gamma}$ to $\frac{2\sigma}{1-\gamma}$ and closed-form sample complexity.** In terms of aggregation and estimation, existing theoretical analyses, such as (Ferns et al., 2011), have primarily focused on the single-MDP setting. Our GBSM not only extends these results to the multi-MDP context but also yields stronger results for the single-MDP case as a direct consequence of our framework. Specifically, our analysis tightens the state aggregation error bound from the existing $\frac{2\sigma (2+\gamma)}{1-\gamma}$ to $\frac{2\sigma}{1-\gamma}$ (Thm. 4.4) and critically, provides a closed-form sample complexity K for estimation (Thm. 4.5), improving upon the prior asymptotic results.
>
> **3. Decoupled error analysis.** Above advancement stems from the core metric properties we establish for GBSM, particularly the inter-MDP triangle inequality. Compared to (Ferns et al. 2011, p. 1701), which necessitates a joint analysis on combined aggregation and estimation error and results in looser bounds and asymptotic complexity, our GBSM property permits a decoupled analysis of different error sources. For example, we can decouple the approximation error $|d^{1-1}-d^{\hat{[1]}-\hat{[1]}}|\leq 2d^{1-\hat{[1]}}$ (continuous MDP → aggregated+estimated MDP) into separated aggregation error $2d^{1-[1]}$ (continuous MDP → aggregated MDP) and estimation error $2d^{[1]-\hat{[1]}}$ (aggregated MDP → aggregated+estimated MDP).
>
> We would improve related description to highlight our contribution and novelty.
>
> > W.4 The results themselves are difficult to interpret …
> >
>
> > Q.1 Can you elaborate more on the construction …
> >
>
> > Q.2 What is the x-axis …
> >
>
> We thank the reviewer for highlighting the need for clarity. The experiments indeed use random Garnet-style MDPs with $|\mathcal{S}| = 20, |\mathcal{A}| = 5$,  and a 50% branching factor. In the aggregation experiments, half of the states are aggregated to a representative state. In the estimation experiments, we introduce a Gaussian noise with a standard deviation ranging from 0.1 to 0.3 into the transition probabilities. For plots, the x-axis represents the experiment index of 100 independent trials. Taking aggregation experiment as an example, the y-axis plots the following three values: the ground-truth aggregation error ($\max|d^{1-1}-d^{[1]-[1]}(s,s)|$, shown as grey dots); GBSM-based theoretical error bound ($2\max d^{1-1_{[\mathcal{S}]}}$, blue dots); and the existing BSM-based bound ( $\frac{2\sigma (2+\gamma)}{1-\gamma}$, red dots). We would extend the current descriptions on coordinates in line 287 to improve the clarity of our results.
>
> > W.4 The empirical evaluation is very limited …
> >
>
> > Q.4 Is it possible to perform any experiments on more interesting MDPs?
> >
>
> We have extended our experiment to a real-world scenario by applying the GBSM to a Digital Twin (DT)-driven DRL task for dynamic resource allocation in a real-world 5G wireless network testbed. In this experiment, the real-world network to be optimized serves as the target MDP, while DTs, as its virtual replicas, function as multiple source MDPs. A DRL policy is trained in each DT and tested for performance in the real network. We employ the proposed aggregation and estimation method to calculate GBSM in real-world, and compare it with policy transfer performance (Detailed in the response to W.2 of reviewer BQkW). Our findings reveal a strong linear correlation between the GBSM distance and the worst-case deployment performance of the policy trained in the DT, not only validating GBSM's predictive capability on transfer regret in a real-world environment but also confirming its aggregation and estimation efficacy in scenarios beyond randomly-generated MDPs. We will incorporate the full results from this real-world experiment into the revised paper, including numerical comparison, a photograph of the Open 5G testbed and corresponding citation (currently omitted to ensure anonymity). While we cannot include plots in rebuttal, we would be happy to provide the raw data (calculated GBSM and corresponding real-world deployment performance) during the discussion period.
>
> # References
>
> [1] Taylor J, Precup D, Panagaden P. Bounding performance loss in approximate MDP homomorphisms. Advances in Neural Information Processing Systems, 2008, 21.
>
> [2] Song J, Gao Y, Wang H, et al. Measuring the distance between finite markov decision processes Proceedings of the 2016 International Conference on Autonomous Agents & Multiagent Systems. 2016: 468-476.
>
> [3] García J, Visús Á, Fernández F. A taxonomy for similarity metrics between markov decision processes. Machine Learning, 2022, 111(11): 4217-4247.
>
> [4] Ferns N, Panangaden P, Precup D. Bisimulation metrics for continuous Markov decision processes. SIAM Journal on Computing, 2011, 40(6): 1662-1714.

---

> > ### Comment · Reviewer_BQkW · 2025-08-05
> >
> > We appreciate the authors' efforts and the work presented in this submission. As we are not specialists in this particular area, we are not in a position to fully assess the technical depth and novelty of the contributions. Therefore, we will primarily rely on the evaluations and judgments of other reviewers with more relevant expertise. Nonetheless, we acknowledge the significance of the problem addressed and the overall clarity of the presentation.

---

> > > ### Comment · Reviewer_xZ9L · 2025-08-06
> > > **Reply to rebuttal**
> > >
> > > I thank the authors for the time and effort they spent on their rebuttal. I will try and address each point individually. I am inclined to maintain my score, and I outline my reasoning below.
> > >
> > > - I thank the authors for being amenable to generalizing the baseline of their metric to both lax-bisimulation and on-policy bisimulation variants. As mentioned in my original review, I think these variants will be of particularly more importance than the original 'max-over actions' considered.
> > >
> > > - I agree that the $2 \sigma / (1 + \gamma)$ bound is an improvement over that of Ferns et al. (2011), but I believe that it is the same bound achieved in Theorem 2 of Kemertas et al. (2021). I also agree with the authors that Theorem 4.5. is novel,
> > > but I overall maintain my opinion that most of the theory has been 'carried over' from previous papers (this isn't a bad thing in itself though -- this is often the case when introducing new yet related concepts.)
> > >
> > > - Thank you for the descriptions of the random MDP experiments!
> > >
> > > - Thank you for the additional experiment and its description. I think that this will strengthen the submission, as it contains more structure to transfer compared to Garnet MDPs.
> > >
> > > As a final note/suggestions to further improve the paper (either for a future submission or for camera-ready depending on acceptance), I would say that I found the first point of the authors' rebuttal convincing for applications of GBSM (games, sim-to-real), and I think that having experiments testing these settings would have greatly improved the paper, and show that the metric can "accomplish what it set out to do", in some sense. For example, continuing on the Pac-Man example of the authors, I think it would be really interesting to see how well the algorithm can transfer a policy from a given Atari game to, for example, the same game where the input channels are passed through a random permutation matrix, to randomize the colours (I think that in order to have non-trivial transfer, you will have to use the on-policy variant of GBSM discussed).

---

> > > > ### Author Response · Authors · 2025-08-07
> > > >
> > > > Thank you for your point to point comment. We are glad that previous concerns 1,3,4 have been addressed, including the AC-mentioned ‘max-over actions’ issue.
> > > >
> > > > On the theoretical contribution, we first thank the reviewer for the acknowledgement of the novelty of Theorem 4.5 on the closed-form sample complexity. Regarding the state aggregation error bound, we wish to clarify that our bound in Theorem 4.4 is different from and tighter than $\frac{2\sigma}{(1-\gamma)}$ in Kemertas et al. (2021).
> > > >
> > > > 1. **In single MDP scenario**, our bound in Theorem 4.4 can be written by $2\sigma'$, where $\sigma'$ is the GBSM distance between the original MDP $\mathcal{M}\_1$ and a specially constructed MDP $\mathcal{M}\_{1_{[\mathcal{S}]}}$,  with $\sigma'=\max_{s\in\mathcal{S}\_1}d^{1-1_{[\mathcal{S}]}}$. While we previously showed $2\sigma'\leq \frac{2\sigma}{(1-\gamma)}$ in line 245 to make an intuitive comparison with Ferns et al. (2011), our bound $2\sigma'$ is actually much tighter than $\frac{2\sigma}{(1-\gamma)}$ when $\gamma$ is large. The numerical validation on Garnet MDPs is provided as follows.
> > > >
> > > > | $\gamma$ | 0.1 | 0.2 | 0.3 | 0.4 | 0.5 | 0.6 | 0.7 | 0.8 | 0.9 |
> > > > | --- | --- | --- | --- | --- | --- | --- | --- | --- | --- |
> > > > | BSM-based $2\sigma/(1-\gamma)$ | 0.111 | 0.181 | 0.255 | 0.389 | 0.669 | 0.818 | 1.311 | 2.374 | 4.159 |
> > > > | GBSM-based $2\sigma'$ | 0.101 | 0.156 | 0.187 | 0.275 | 0.390 | 0.455 | 0.618 | 1.078 | 1.536 |
> > > >
> > > > 2. More importantly, the primary contribution of our work is in **multi-MDP analysis**, where our framework offers substantial advantages over BSM-based single-MDP methods. Existing approaches would necessitate constructing a disjoint union of the MDPs, $\mathcal{M}\_{1\cup 2}$, with isolated transition probabilities, regarding it as one large MDP, and calculating BSM on it (Phillips, 2006; Ferns et al., 2011). Applying the bound from Kemertas et al. yields a bound of $\frac{2\sigma_{1\cup 2}}{(1-\gamma) }= \frac{2\max(\sigma_1,\sigma_2)}{(1-\gamma)}$. In contrast, our GBSM is directly computed between these two MDP, yeilding a tighter bound of $\sigma'\_1+\sigma'\_2\leq\frac{(\sigma_1+\sigma_2)}{(1-\gamma)}$. This is especially true in practical transfer learning scenarios like sim-to-real, where the source MDP (simulator) $\mathcal{M}\_1$ is a fully known environment where the state space partition could be very fine-grained ($\sigma_1\ll\sigma_2$) or even minimum ($\sigma_1=0$). In this multi-MDP context, the existing bound in Kemertas et al. (2021) would be further looser than our bound in Theorem 4.4.
> > > >
> > > > 3. Meanwhile, calculating BSM on this disjoint union, $\mathcal{M}\_{1\cup 2}$, would render a **significant computational complexity** scaling with $|\mathcal{S}\_1+\mathcal{S}\_2|^2$. In contrast, our GBSM is directly computing between $\mathcal{M}\_1$ and $\mathcal{M}\_2$, with an reduced complexity scaling with $|\mathcal{S}\_1|\cdot|\mathcal{S}\_2|$.
> > > >
> > > > Finally, regarding your comment that our theory is "carried over", our work is indeed a principled extension of BSM to the multi-MDP domain, but this process is highly non-trivial. Unlike BSM, our GBSM is no longer an inherent pseudometric on a single state space, requiring us to re-establish fundamental metric properties from scratch. For example, proving the inter-MDP triangle inequality (Theorem 3.5) required a novel proof involving an extra introduced MDP $\mathcal{M}\_3$, and a joint transportation plan constructed via the Gluing Lemma. These rigorous theoretical extension is precisely what enables the key advantages of our work, e.g., tighter performance bounds, reduced computational complexity, and the first closed-form sample complexity, etc. These contributions provide a solid theoretical foundation for computing similarities between multiple MDPs, moving beyond the primarily empirical approaches, as shown in (Song et al., 2016) and highlighted in the survey (García et al., 2022).
> > > >
> > > > We thank the reviewer again for their detailed and constructive suggestions. We will replace the baseline of Ferns et al. (2011) by Kemertas et al. (2021) in the revised version, and highlight the exceptional contribution of our work compared to the existing literature. We hope this clarifies the novelty and significance of our contributions and are happy to provide any further details.

---

> ### Comment · Area_Chair_gGAL · 2025-08-03
>
> Dear reviewer,
>
> Thank you for your efforts so far in reviewing this paper. The authors have answered your questions. Would you please check their reply and let them and the other reviewers know what you think about their response?
>
> Specifically, one of your comments attracted my attention: "The variant of the bisimulation metric used is the original (which takes $\max$ over actions), which has been shown to have a number of drawbacks (lack of connection to non-optimal policies, biased estimation, etc)."
>
> I would also very much like to know the opinion of other reviewers too. As I understand, this issue affects the meaningfulness of the results in the paper.
>
>
> Thank you,
> Area Chair

---

> ### Author Response · Authors · 2025-08-04
>
> Thank you to the Area Chair for raising this crucial point. We chose BSM in our main paper for its established role as a clear baseline, but a key advantage of our GBSM is its compatibility, which allows adaptation to more robust, modern variants.
>
> As detailed in our response to Reviewer xZ9L (under **Lax GBSM**), our framework can be readily adapted for lax BSM (Taylor et al., 2008), enabling a $\max_a \min_{a'}$ comparison instead of the standard $\max_{a,a'=a}$.
>
> Also, it is compatible with **on-policy BSM** (Castro, 2020) by rewritten (2) in our paper to $d^{1-2}\_\pi(s,s')=|R_1^\pi(s)-R_2^\pi(s')| +\gamma W_1(\mathbb{P}\_1^\pi(\cdot|s), \mathbb{P}\_2^\pi(\cdot|s') ;d^{1-2}\_\pi)$, where $R_{.}^\pi(s)=\sum_a \pi(a|s)R_{.}(s,a)$ and $\mathbb{P}\_.^\pi(\cdot|s)=\sum_a \pi(a|s)\sum_{\tilde{s}}\mathbb{P}\_.(\tilde{s}|s,a)$ are averaged reward and transition probabilities for a **non-optimal policy** $\pi$. Crucially, our core theoretical properties are preserved: the value difference bound (Thm. 3.3) now applies to the on-policy value function by $|V^\pi_1(s)-V^\pi_2(s')|\leq d_\pi^{1-2}(s,s')$. Then metric properties Thms. 3.4-3.5 follow directly, and $d^{1-2}\_\pi(s,s')$ is bounded by a similarly adapted on-policy TV-based metric $d^{1-2}\_{\text{TV},\pi}(s,s)=\{|R_1^\pi(s)-R_2^\pi(s)\big|+\frac{\gamma \bar{R}}{1-\gamma} \text{TV}(\mathbb{P}\_1^\pi(\cdot|s), \mathbb{P}\_2^\pi(\cdot|s) )\}$
>
> The fact that our GBSM can be seamlessly extended to on-policy and lax variants demonstrates its generality. While our experiments use the standard metric for clear comparison with prior work, the underlying contribution is a flexible framework. We believe this adaptability significantly strengthens our contribution and addresses the concerns about the meaningfulness of our results. We will clarify this in the final version of the paper.
>
> Thank you again for your insightful feedback. We are happy to answer any further questions.

---

### Decision · Program_Chairs · 2025-09-17

**Decision:**

Accept (poster)

**Comment:**

The paper generalizes the bisimulation metric (BSM) from a metric on the states of an MDP to a metric between the states of two MDPs, hence extending BSM to multi-MDP scenario.
The paper proves several properties of this generalized BSM, including relating the difference between the optimal value of two MDPs to their generalized BSM metric (Theorem 3.3), performance bound on policy transfer between MDPs (Theorem 4.1), and a sampling-based estimation of the error between the actual metric between two MDPs and the one learned using samples (Theorem 4.5). The paper provides some simple empirical illustrations too.

We have one accept (5), two borderline accepts (4), and one reject (2). The positive reviewers have low confidence (2 or 1), and the negative reviewer has high confidence (4).

All reviewers believe that the theory of this work is solid and rigorous. The main negative issues, raised by Reviewer xZ9L, are that (1) theoretical work are straightforward adaptation of existing results and (2) the suitability of the variant of BSM that is generalized.

Regarding point (2): The authors have provided the sketch of how different variants of BSM (lax bisumulation and on-policy bisimulation) can be generalized.

Regarding the novelty (point 1), the main issue is not whether the results are already existing or not, as the previous similar theoretical results are for a single MDP and these results are for multi-MDP after all. The issue is whether the generalization of the existing results from a single MDP to multi-MDP case is easy or requires sophisticated mathematical techniques. I do not comment on the required level of sophistication, as I have not read the proofs. My stance, however, is that the more important point is the novelty of the problem setting, and not the novelty of the mathematical techniques required to prove results for the new problem setting. Since the problem setting is novel (metric between two MDPs), the paper does not lack novelty.

Overall, this is a reasonably good paper and should be accepted, despite its relatively lower average score. I ask the authors to consider the comments of all reviewers to improve their work. Specifically, I ask them to fully develop the proof of generalization of other notions of BSM. I also encourage them to include more practically "interesting" experiments to showcase the usefulness of their framework.